# Dimensionality Mismatch Between Brains and Artificial Neural Networks

**Santiago Galella**
FIAS & Institute of Computer Science
Goethe University Frankfurt
galella@fias.uni-frankfurt.de

**Maren Wehrheim**[*]
Mila & Department of Biology
York University
maren.wehrheim@mila.quebec

**Matthias Kaschube**[*]
FIAS & Institute of Computer Science
Goethe University Frankfurt
kaschube@fias.uni-frankfurt.de

## Abstract

Biological and artificial vision systems both rely on hierarchical architectures, yet it remains unclear how their representational geometry evolves across processing stages, and what functional consequences may arise from potential differences. In this work, we systematically quantify and compare the linear and nonlinear dimensionality of human brain activity (fMRI) and artificial neural networks (ANNs) during natural image viewing. In the human ventral visual stream, both dimensionality measures increase along the visual hierarchy, supporting the emergence of semantic and abstract representations. For linear dimensionality, most ANNs show a similar increase, but only for pooled features, emphasizing the importance of appropriate feature readouts in brain–model comparisons. In contrast, nonlinear dimensionality shows a collapse in the later layers of ANNs, pointing at a mismatch in representational geometry between the human and artificial visual systems. This mismatch may have functional consequences: while high-dimensional brain representations support flexible generalization to abstract features, ANNs appear to lose this capacity in later layers, where their representations become overly compressed. Overall, our findings propose dimensionality alignment as a benchmark for building more flexible and biologically grounded vision models.

## 1 Introduction

The geometry of functional representations has been linked to core computational properties of both biological and artificial systems, including accuracy, flexibility, and generalization [e.g., 1, 12, 21, 25, 43, 61, 67]. One key geometric property is representational dimensionality, which quantifies how neural activity is embedded in low- or high-dimensional subspaces in response to different stimuli or tasks [2, 11]. Various metrics have been proposed to quantify the representational dimensionality [e.g., 13, 11, 67, 58]. Linear measures such as the effective dimensionality describe how variance is spread globally, while nonlinear estimates (intrinsic dimensionality) capture the number of degrees of freedom needed to locally approximate the data manifold. Understanding how these geometric properties evolve along the visual hierarchy is critical for identifying functional distinctions across processing stages and for aligning representational principles between brains and models.

In machine learning, the manifold hypothesis posits that high-dimensional inputs lie on low-dimensional manifolds embedded in ambient space [6]. ANNs leverage this by compressing task-

---

[*]Joint supervision

39th Conference on Neural Information Processing Systems (NeurIPS 2025).

relevant information into compact, discriminative subspaces [4]. Prior work has shown that neural networks typically exhibit an expansion–compression pattern of dimensionality across depth: early layers increase dimensionality to represent diverse features, while deeper layers compress representations to facilitate categorization [1, 61, 67]. While such compression is linked to improved classification performance [23], high intrinsic dimensionality has been associated with improved generalization to novel categories [21], raising the possibility that over-compression may impair semantic abstraction.

In the brain, it remains unclear how representational geometry changes along the processing hierarchy. While early sensory representations are traditionally considered to occupy low-dimensional subspaces [26, 80, 40], recent work using naturalistic conditions reveals that neural activity often resides on high-dimensional manifolds across the whole brain, even in early sensory areas [3, 10, 32, 58, 62, 70]. Distinct dimensions were shown to encode not only sensory input but also abstract semantic and linguistic information [39, 38, 14]. Moreover, several studies have produced diverging hypotheses about how representational dimensionality evolves along the visual hierarchy. While studies in mice and zebrafish found a continuous expansion of the dimensionality [17, 77], others reported a compression-expansion pattern in macaques [67]. These contrasting patterns highlight the open question of how the human brain organizes information across processing stages in the visual cortex and how this compares to state-of-the-art ANN representations.

Here, we investigate commonalities and differences in representational geometry between biological and artificial vision systems, and examine possible functional consequences of such differences for visual abstraction and generalization. We use the THINGS dataset, which contains fMRI data from humans viewing naturalistic images spanning multiple object categories, along with human ratings of these images on abstract attributes such as 'natural' and 'heavy' [34, 69]. We compute both linear (effective) and nonlinear (intrinsic) dimensionality across the human visual hierarchy, taking cortical distance from the central visual field representation in the primary visual cortex as a proxy for position within the hierarchy, and directly compare these profiles to those we obtain for state-of-the-art vision models.

We find that in the human ventral stream, both effective and intrinsic dimensionality increase along the hierarchy, suggesting an expansion of representational capacity to support semantic abstraction. In vision models, we find a consistent increase in effective dimensionality, but only if we assess representations in these ANNs using global average pooled features, possibly because this may better reflect the coarse spatial resolution of fMRI. In contrast, ANNs exhibit a collapse in intrinsic dimensionality in deeper layers, even when using pooled features, revealing a fundamental mismatch in representational geometry between brains and models. This mismatch suggests functional implications: we find that in the brain, high-dimensional regions are better at encoding abstract, behavioral attributes, whereas in ANNs, despite their increase in categorization, the deeper layers that display the collapse in intrinsic dimensionality tend to show a decline in performance on the abstract concepts. These results suggest that nonlinear dimensionality is an important characteristic of representational geometry and that its collapse in ANNs could be associated with a limited generalization to abstract concepts.

Overall, our contributions are as follows:

- We demonstrate that global average pooled ANN features best align with fMRI-derived linear geometry, highlighting the importance of population-level readouts in model–fMRI comparisons.

- We reveal a functional mismatch in representational geometry between brains and ANNs: while both effective and intrinsic dimensionality increase along the human ventral stream, ANNs exhibit a marked compression of intrinsic dimensionality in their deeper layers.

- We link this mismatch to function by showing that higher-dimensional representations in the brain are accompanied by a superior generalization to abstract, behaviorally rated concepts, whereas nonlinear compression in ANNs tends to correlate with lower generalization abilities despite high categorization performance.

Our findings suggest that geometric properties such as intrinsic dimensionality carry functional significance, and aligning these between ANNs and the brain may be key to improving model generalization and interpretability.

## 2 Methods

### 2.1 Dimensionality

Dimensionality refers to the number of independent dimensions required to describe the data. Here, we define dimensionality in terms of effective dimensionality (ED), a linear measure for dimensionality, and intrinsic dimensionality (ID), which is a widely used nonlinear measure of dimensionality.

The *effective dimensionality (ED)*, also known as *participation ratio*, estimates the number of dimensions that capture most of the variance of the data [27, 18, 21]. For a data matrix $A \in \mathbb{R}^{N \times M}$, with $N$ images and $M$ features (number of model units or vertices), the ED is calculated as:

$$\text{ED} = \frac{(\sum_{i=1}^{n} \lambda_i)^2}{\sum_{i=1}^{n} \lambda_i^2} \, , \tag{1}$$

where $\lambda_i$ are the eigenvalues of the covariance matrix of $A$, and $n = \min(N, M)$.

The *intrinsic dimensionality (ID)* estimates the dimensions of the manifold in which the data is embedded [47, 18], i.e., the minimal number of dimensions that capture the data distribution locally. It can be estimated via maximum-likelihood (MLE) by modeling neighbor counts within a small radius around each data point as a homogeneous Poisson process and relating their expected number to the volume of a d-dimensional hypersphere [47, 51]:

$$\text{ID} = \left[ \frac{1}{n(k-1)} \sum_{i=1}^{n} \sum_{j=1}^{k-1} \log \frac{T_k(x_i)}{T_j(x_i)} \right]^{-1} , \tag{2}$$

where $n$ is the number of samples in the dataset, $k$ is the $k^{th}$ nearest neighbor of the data point, and $T_k(x_i)$ is the Euclidean distance from point $x_i$ to its $k$-th nearest neighbor. Here, we set $k$ to be 20.

Hence, whereas ED reveals the effective number of global variance components, ID reflects local structures on complex, curved manifolds that may not be apparent in linear projections. Note that ID can be smaller than ED and vice versa. For instance, a one-dimensional helix spiraling through three-dimensional space has ID = 1 but can exhibit ED close to 3 if its variance is spread roughly evenly across the x, y, and z axes; by contrast, a filled 3-ball that is stretched along one axis can have ED close to 1 despite its true intrinsic dimension being ID = 3.

Note that both of our dimensionality measures are unsupervised, hence they do not require explicit [13] or random categorical [58] information.

### 2.2 fMRI data

We used the THINGS fMRI dataset [34] to study dimensionality of brain activity while viewing natural images. In the THINGS fMRI dataset, three participants (S1-S3) completed 12 scanning sessions each, while viewing a total of 9,840 images. From these, 8,640 were unique image repetitions of 720 object categories, and the remaining 1,200 images comprised 100 repeats. All stimuli were selected from the THINGS database, which contains 26,107 distinct images spanning 1,854 object categories [35].

Additionally, we use the BOLD5000 dataset, which contains fMRI data of four different subjects (CSI1-CSI4) while viewing natural images of objects (from ImageNet-1K and COCO datasets) and scenes (from Scenes dataset) [7]. The total number of images observed by the subjects were 5234, from which 4916 were unique (for the fourth subject, the total number of images was 3108).

For both datasets, we performed surface-based analysis by projecting the beta values linearly from functional volumes onto the cortical mid-thickness surface, between the pial and white matter surfaces, to minimize noise from superficial and deep cortical layers [9]. We use the HCP parcellation to subdivide visual cortex into five large processing stages: the primary visual area, early visual area, dorsal stream visual area, ventral stream visual area, and MT+ Complex and Neighboring visual areas, which we term the lateral stream; each larger area is then further subdivided into smaller subregions [see A2.2 30]. In all analyses, we used a searchlight approach where a fixed-size spherical (or surface-based) neighborhood mask of contiguous vertices is moved across the cortex. For each

vertex, we selected a neighborhood of the 50 closest vertices (including the vertex itself). On average, these neighborhoods have a maximal distance close to 7 mm. To compute cortical distances between brain areas, we first identify the vertex for each area that has the minimal sum of distances to all other vertices in that area (the surface centroid). We then measure the distance from the V1 centroid in primary visual cortex to the centroids of the other visual areas.

We calculate local dimensionality using searchlight analysis. While this allows for a local dimensionality estimate, it also ensures an upper bound of dimensionality across areas, preventing larger regions from artificially inflating dimensionality estimates. We obtain area-specific dimensionality estimates by averaging across all local dimensionality estimates in a specific area. Hence, our approach ensures comparability across cortical areas, while avoiding biases introduced by varying area sizes.

We repeat all analyses using searchlight neighborhoods of 25 (5 mm) and 100 (10 mm) nearest neighbors (Figs. A14, A15). Additional information on the methods and results for the different subjects are included in Appendix A4.

### 2.3 Artificial Neural Network Models

In addition to the fMRI data, we study artificial neural networks (ANNs), as potential models of visual processing in the brain. We analyze 36 different ANNs, including convolutional neural networks and vision transformers trained with supervised or self-supervised objectives as well as autoencoders to test the influence of unsupervised objectives (see Table A5 for a list of all models and more details on training objectives). Since our study also aims to bridge biological and artificial representations, we further include models inspired by biological constraints, such as CORNet-S [45], which is explicitly designed to approximate the primate ventral stream hierarchy. Moreover, to evaluate the role of training data in shaping representational structure, we also include models trained on more ecologically valid datasets, such as Ecoset [54], which mimics human object category statistics, and EgoVLP [48], which learns from egocentric, first-person visual input. This range of models enables us to ask not only how dimensionality differs between ANN architectures, training data, and objective functions, but also whether models with more biologically plausible features (in terms of architecture or data) show greater alignment with the hierarchical dimensionality profiles observed in human ventral visual cortex.

For all models, we extract convolutional feature activations after ReLU activations (GELU for transformer-based models) for unique THINGS images that were viewed by the subjects in the fMRI scanner. We then apply global average pooling (GAP) to each feature map to focus on conditionally independent image-based features and remove spatial variance [21]. To generate comparable results across layers, we repeatedly subsample 50 GAP features. Additionally, we include analyses using i) random projections, that compress the feature dimensions onto 50 random sparse vectors while preserving relative distances between samples [67], and ii) 50 randomly subsampled unit activations (without GAP) to preserve each unit's tuning properties [41] in Appendix A7.1. We define the depth of a layer in the model's processing hierarchy by dividing the layer's position by the total number of all processing steps in the model [1, 67].

## 3 Results

### 3.1 Diverse dimensionality throughout cortex

We first examine how representational dimensionality varies across the human visual cortex by analyzing whole-brain fMRI data from the THINGS database, where participants passively viewed thousands of naturalistic object images. Specifically, we calculate the *local* dimensionality for all vertices across visual cortex using a surface-based searchlight approach [9]. We computed both effective dimensionality (ED) and intrinsic dimensionality (ID) of the vertex-wise responses as complementary measures to quantify the geometry of fMRI representations: While ED captures how the variance is distributed across linear response dimensions, ID estimates the underlying degrees of freedom required to describe the nonlinear manifold the data resides in. We found that dimensionality is highly diverse across regions in the visual cortex (Fig. 1a). Notably, we find that all visual areas show a lower dimensionality compared to areas in the ventral stream (Fig. 1b). Hence, the higher dimensionality in these regions might be associated with their involvement in handling rich, complex

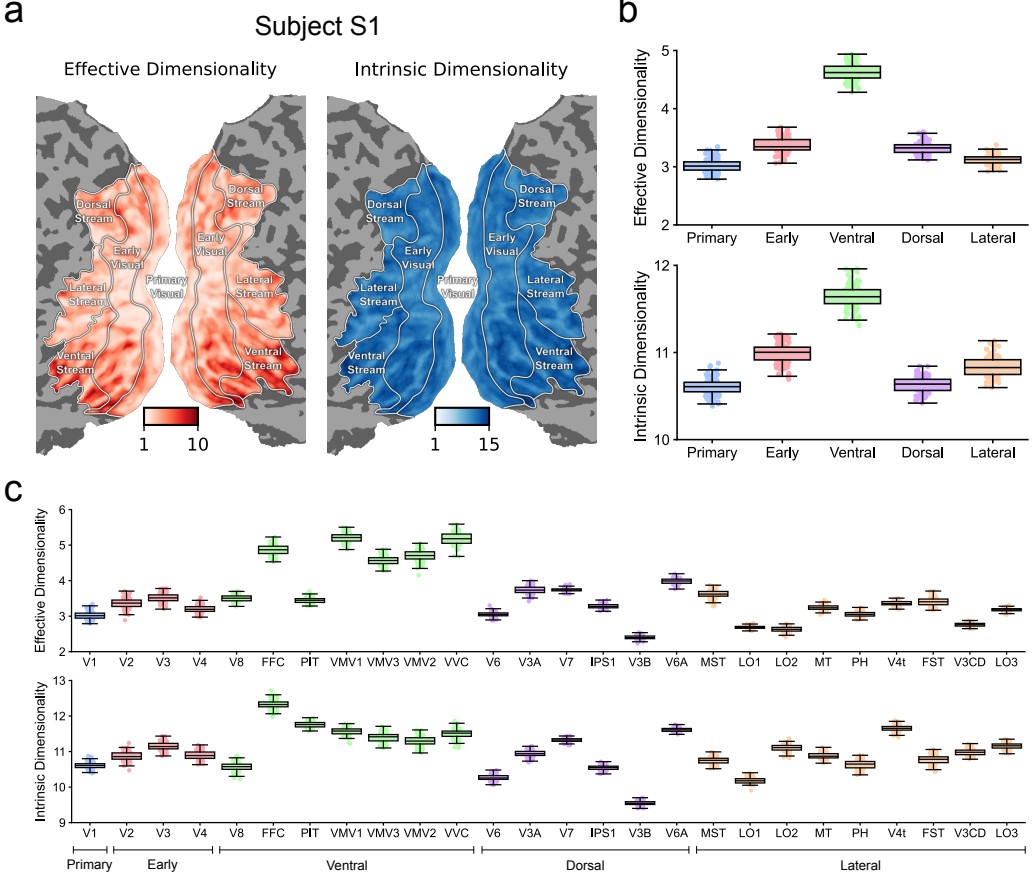

Figure 1: **Dimensionality varies across cortex. a**) Effective (ED) and intrinsic (ID) dimensionality across the visual cortex for subject S1 from THINGS fMRI dataset. Dimensionality was computed using a searchlight approach with 50 vertices (see Methods). Dimensionality averaged across larger brain regions (**b**) and smaller sub-regions (**c**). The boxes show the quartiles of dimensionality across searchlight discs in an area.

representations of static objects, and not a feature of the processing hierarchy as the dorsal stream's more specialized role in action and movement exhibits a low dimensionality.

Zooming into these coarse regions reveals dimensionality patterns to be spatially specific and reproducible across finer cortical areas, indicating a stable organizational structure in how visual information is represented across the brain (Fig. 1c). Specifically, regions like the Fusiform Face Complex (FFC), responsible for face processing, the Ventromedial Visual Area (VMV), associated with spatial scene processing, and the Ventral Visual Complex (VVC), commonly known for object recognition, show high dimensionality. The higher dimensionality observed in these regions may reflect a greater differentiation between various object categories and more abstract features compared to other visual areas. Despite potential variations in the processing of visual inputs between hemispheres, both ED and ID were highly correlated across hemispheres (Figs. A4, A5). Also, ED and ID in fMRI data were strongly correlated (Figs. A6, A7), suggesting common underlying changes in the geometric properties of the BOLD response along the visual processing hierarchy. Overall, these differences in dimensionality for different areas are consistent across subjects (Fig. A8). Early visual areas exhibit a local retinotopic organization, which may reduce dimensionality estimates due to spatial correlations among neighboring vertices. To control for this, we repeated the analysis using randomly sampled, non-contiguous vertices within each region. Despite expected shifts in absolute dimensionality values, the hierarchical trend across regions remained robust (Fig. A17).

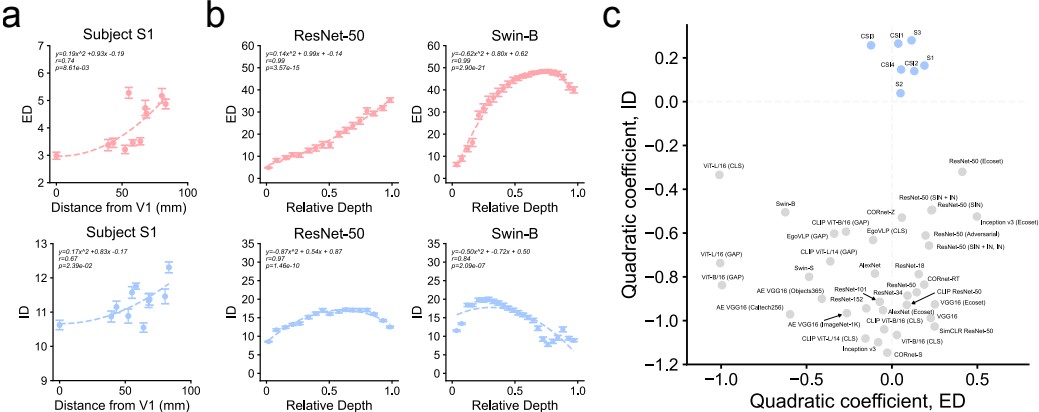

Figure 2: **Hierarchical changes in dimensionality reveal a mismatch between brains and ANNs a**) Effective (ED, top) and intrinsic (ID, bottom) dimensionality for human fMRI data along the ventral visual stream. Values for different areas (using the data from Fig. 1c) are plotted as a function of geodesic distance along the cortical surface from the primary visual cortex (see Methods). The error bars show the standard deviation across 100 bootstrapped samples from the results of 100 randomly selected vertices per region. **b**) Dimensionalities for two example ANNs (ResNet50, ViT) with global average pooled features. Relative depth indicates the normalized index of the network layer. The error bars represent the standard deviation of the dimensionality calculated for 100 random subsets of 50 globally pooled features per layer. **c**) Quadratic coefficients of second-order polynomial fit for ED and ID along the visual hierarchy in humans (blue dots) and models (gray dots).

## 3.2 Diverging dimensionality trends highlight fundamental differences between brains and ANNs

### 3.2.1 Dimensionality increases along the ventral stream

To quantify how dimensionality changes across the ventral visual stream, we examined how both ED and ID evolve as a function of distance from primary visual cortex (V1). Specifically, we defined visual hierarchy depth as the geodesic distance from the center of V1 to each region of interest (see Appendix A6.2 for other measures of visual hierarchy depth). We found that both ED and ID increase with depth (Fig. 2a, top and center), often in a superlinear manner. This increase might correspond to a functional expansion of the brain's representation space to more diverse and selective tuning, reflecting untangled, abstract feature dimensions.

### 3.2.2 Global-average pooling aligns the linear geometry of artificial networks with the brain

Contrary to what we observe in the brain, previous work has consistently reported an expansion–compression pattern of dimensionality in ANNs, characterized by an initial increase followed by a decrease across layers [e.g., 1, 61, 67]. To investigate whether this pattern is robust and holds under more naturalistic conditions, we analyze ED using the same large-scale image dataset (THINGS) as for fMRI in 36 different pretrained ANNs, varying in architecture and training objective. When ED is computed using randomly projected or subsampled units, we replicate the previously observed increase–decrease trend across all ANN layers (see Fig. A18). In contrast, when we compute ED using global average pooled (GAP) feature maps, which emphasize image-level over spatial variance, we observe an increase in dimensionality with depth in most models (Fig. 2b top row ResNet-50, Fig. A11), mirroring the trend observed in the human ventral stream. GAP preserves high-level semantic content and feature identity while discarding spatial detail, potentially paralleling the spatial integration performed by fMRI voxels across millimeter-scale cortical areas. We confirm that this increase in ED emerges only during training, and is not present in randomly initialized networks. However, for ViT-based models we again observe a decrease in ED in later layers (Fig. 2b top row Swin-B) indicating a fundamental difference between geometric representations in the ViT architecture compared to CNNs and the brain. Overall, these results highlight a critical consideration

for cross-domain alignment: coarse-grained, pooled features may more accurately reflect population-level signals in the brain, and thus serve as a better proxy for fMRI recordings in representational geometry analyses.

### 3.2.3 Divergent intrinsic geometry in brains and artificial networks

Having observed that global average pooling can resolve the mismatch in effective dimensionality between the brain and many ANN architectures, we next ask whether the same holds for ID which is more sensitive to the local, nonlinear structure of the representational manifold. As before, we apply global average pooling across feature maps to reduce spatial variance. However, in contrast to the brain, where ID increases along the ventral stream, we find that in all ANNs, ID follows the canonical hunchback shape defined by an initial increase followed by a continuous decrease (Fig. 2b bottom row, Fig. A12) consistent with previous findings [1, 61, 67]. The dimensionality collapse is not limited to task-optimized layers but persists in those most aligned with late-stage visual areas, including IT (Fig. A20), revealing a fundamental geometric mismatch between biological and artificial representations that cannot be resolved by pooling alone.

To systematically compare representational trends, in particular the strength of dimensionality collapse, we fit a second-order polynomial (i.e., a model including both a linear and quadratic term) to the progression of both ED and ID along the visual hierarchy in brains and ANNs. For ED, the coefficients of the quadratic term obtained for the human subjects overlap with those of the ANNs, suggesting overall comparable trends (Fig. 2d). However, for ID, we observe a clear divergence: whereas all subjects show a positive quadratic coefficient, indicating an upward curved trend where high-level regions maintain a higher ID than other regions, all ANNs show a negative quadratic coefficient, reflecting the mid-processing peak and sharp decline (collapse) in ID in their deepest layers. This pattern persists across all tested ANNs, including those with brain-inspired architectures (e.g., CORNet-S [45]), models trained on more ecologically valid datasets such as Ecoset [54], and those trained with egocentric visual experiences designed to approximate human perception [48]. Despite these biologically motivated design choices, all exhibit a similar late-stage decline in intrinsic dimensionality. Taken together, these findings highlight the relevance of studying the nonlinear representational structure by revealing a fundamental difference in how biological and artificial systems represent information towards the higher levels of their processing hierarchies.

In summary, we observe that ED increases along the processing hierarchy in both fMRI data and most ANNs, reflecting a growing diversity of linear subspaces. However, while fMRI representations retain high ID in higher-order regions, ANNs exhibit a sharp reduction in ID in their final layers. We hypothesize that this reflects task-specific compression, where the network reduces representational complexity to optimize performance on a single objective. To test this possibility, we next examine how different types of properties, ranging from visual features to abstract semantics, can be decoded across brain and model hierarchies.

### 3.3 Low-level image features are represented in early layers

Having shown that dimensionality trends differ in brains and ANNs in higher-order regions, we next examine what type of information is actually represented across the processing hierarchy in both systems. While dimensionality metrics describe the structure of representational space, they do not reveal its content. To address this, we perform decoding analyses to test whether regions differ in their ability to represent low-level visual features as well as semantic and categorical content. Specifically, for fMRI data, we first decode the presented stimuli using leave-one-session out cross validation using logistic regression. In line with prior work [e.g., 36], decoding accuracy is highest in primary and early visual areas, especially in regions with low eccentricity (Fig. 4a, left). To assess the representation of low-level features more directly, we extract and predict the first principal components of the images (see Supplementary Material), which capture contrast, spatial frequency, and color from the fMRI data using ridge regression. These low-level features are again predominantly represented in early visual areas (Fig. 3a, right), except color (component 4), which is more relevant in later regions. Since stimulus decoding is not applicable in deterministic ANNs, we only evaluate how well low-level image features are encoded across layers. In line with the brain, we find that early ANN layers most strongly encode these features and the performance continuously drops with layer depth (Fig. 3b). A similar drop in performance for the color component highlights a mismatch in how color information is encoded in brains versus models. Together, these results confirm that early stages

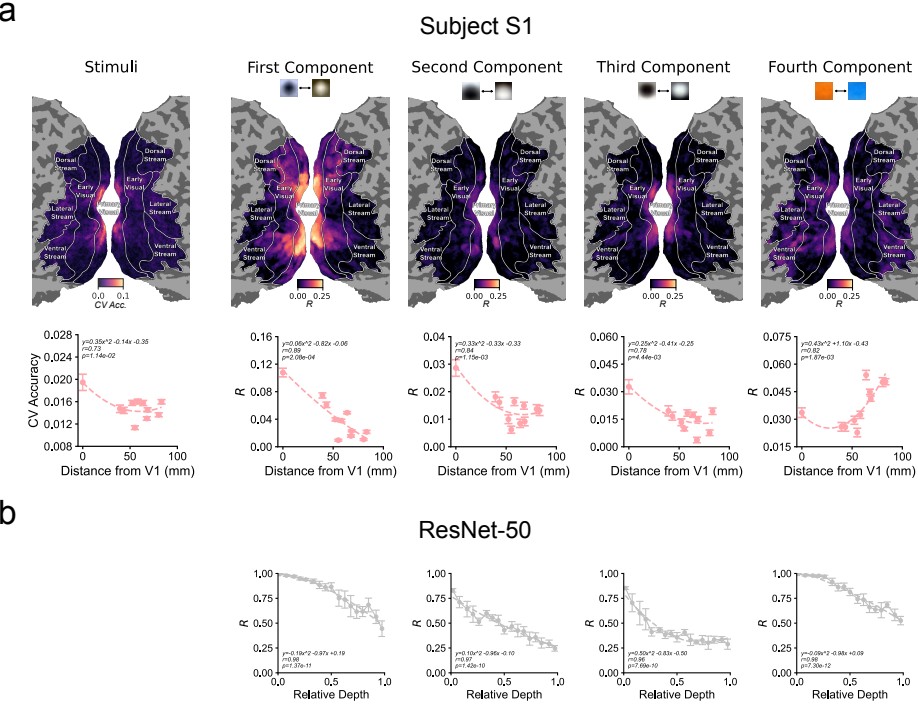

Figure 3: **Decoding low-level features (image components) from brains and ANNs**. Decoding stimuli (i.e. the image, left) and image components (2nd left to right) of images for subject S1 from THINGS fMRI dataset (**a**) and ANNs (**b**). For the fMRI data, the scatter plots indicate the mean and standard deviation obtained by repeatedly sampling the results (100 times) for 100 vertices per region across the early, primary, and ventral streams (11 regions; see Fig. 1c). Cortical distance is measured as in Fig. 2a. For ANNs, the plots show the mean and standard deviation across 10 train-test splits (80/20), each using a subset of 50 averaged feature maps.

in both brains and models reliably encode low-level visual features, supporting general-purpose, high-fidelity representations of sensory input.

### 3.4 Decoding performance of abstract concepts aligns with nonlinear dimensionality in the brain and artificial networks

Having established that early stages in brains and ANNs represent low-level image features, we next assess whether higher-order regions encode more abstract, semantically meaningful information and whether this aligns with the observed differences in representational geometry.

In fMRI, we first decode object categories, and find that decoding performance increases continuously along the ventral stream, with category-level information being strongest in higher-order ventral areas, consistent with their established role in object recognition (Fig. 4a,b left). To move beyond categorical labels, we next predict human-rated semantic attributes of each image, such as heavy or natural, from brain activity [69]. Again, we find high prediction performance in ventral but also lateral regions (Fig. 4a,b right) with a continuous increase along the ventral stream, highlighting that high-level areas flexibly represent abstract features that generalize beyond categorization.

In contrast, while ANNs generally improve in categorization accuracy across layers, for abstract concepts, this performance trend often plateaus or even declines in deeper layers (Fig. 4c). This is reminiscent of the collapse in ID (Fig. 2), and we quantify this trend, as before, by fitting a second-order polynomial to the change in prediction performance. For most ANNs we observe a negative quadratic coefficient for abstract feature regression (Fig. 4d), confirming that indeed most ANNs show a performance decline in higher layers. Additionally, we find the quadratic coefficients for ID

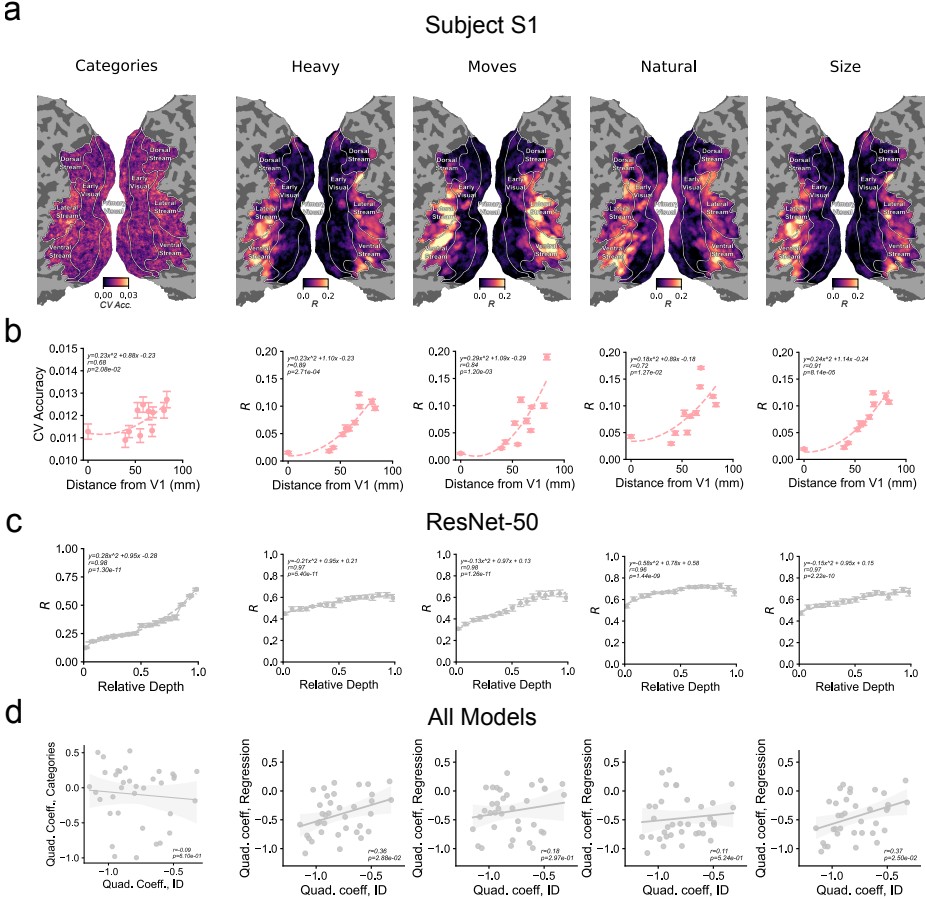

Figure 4: **Decoding high-level features (abstract attributes) from brains and ANNs**. Decoding categories (left) and abstract attributes from images (2nd left to right) for subject S1 from THINGS fMRI dataset (**a, b**) and ANNs (**c**). For the fMRI data, the scatter plots indicate the mean and standard deviation when repeatedly sampling the results (100 times) for 100 vertices per region across the early, primary, and ventral streams. Cortical distance is measured as in Fig. 2. For ANNs, the plots show the mean and standard deviation across 10 train-test splits (80/20), each using a subset of 50 averaged feature maps. **d**) The coefficients of the quadratic term of a second-order polynomial fit for ID and categorization (left) and abstract feature decoding performance (right) along the visual hierarchy in the ANNs from Fig. 2.

to correlate positively with those of several abstract feature decodings, but not with categorization (Fig. 4d), potentially indicating a connection between the strength of ID collapse and the inability to form ordered abstract representations across models. This suggests that late-stage compression in ANNs supports task-specific optimization at the cost of rich, semantically grounded representations that the model was not explicitly trained for.

Our results, thus, suggest a functional divergence: while ANNs compress their representations to optimize for narrow tasks, the brain preserves high-dimensional geometry in higher-order areas, enabling semantic abstraction and behavioral generalization. This underscores the value of aligning not only representations but also geometric properties when building biologically inspired models.

## 4  Discussion

This work highlights a fundamental mismatch in representational geometry between the human visual cortex and ANNs. We report a continuous increase in both effective and intrinsic dimensionality

along the human ventral stream, while ANNs exhibit a collapse of intrinsic dimensionality in deeper layers that best align with higher visual areas in the brain. This mismatch appears functionally meaningful: high intrinsic dimensionality in the brain is accompanied by high generalization to abstract, behaviorally rated features, whereas in most ANNs, abstraction performance declines in the deeper layers, despite improved categorization, and, as a trend, this decline seems stronger in those models that also show a more pronounced ID collapse.

How ANNs align with the brain is an active field of research, see e.g. [15, 74, 41, 55, 57, 76, 79]. Here, we show that global average pooled features in ANNs align more closely with the population-level geometry observed in fMRI than unit-level features or random projections. However, these pooled representations may not capture fine-grained neural dynamics and are therefore less suitable for comparison with electrophysiological recordings, where single-unit activations may provide a closer match. While the brain-alignment of effective dimensionality relies on pooled features, the drop in intrinsic dimensionality remains across all preprocessing methods employed. This drop also appears in self-supervised contrastive models, suggesting that dimensionality collapse can emerge independently of supervised objectives. These results point to a deeper difference in how abstraction is supported across systems: Whereas the brain may promote high-dimensional structure to support semantic flexibility, ANNs tend to optimize for compression, potentially limiting their generalization.

Notably, the use of high-dimensional manifolds in the brain may facilitate flexible readouts in downstream areas, as embeddings into higher-dimensional spaces can make complex classification problems more likely to be linearly separable [16, 25]. While nonlinear dimensionality reflects intrinsic representational degrees of freedom, linear decodability assesses how readily features can be linearly extracted—an approach also used to estimate manifold capacity [13, 58]. The observed drop in nonlinear dimensionality in deeper ANN layers, which appears to correlate with a reduced linear decodability for some abstract features, suggests that representations become increasingly compressed along task-relevant dimensions, limiting access to non-task-related features.

Additionally, it would be interesting to study how the reported dimensionality mismatch emerges during training and whether deeper ANN layers better align with higher-order cortical regions (e.g. prefrontal cortex) [60]. These insights could inform model design strategies that avoid premature compression and better preserve the geometry needed for generalization [29]. At the same time, previous work reports that intermediate layers maximize alignment with measured brain responses [72, 50] and can exhibit stronger generalization [75, 66]. This is consistent with findings linking higher dimensionality to improved generalization [21]. In our results, intrinsic dimensionality also peaks at intermediate layers, suggesting a link between this regime and generalization performance.

Together, our results suggest that the increase in representational dimensionality is not only an architectural feature but a functional signature of abstraction and generalization in vision systems. By uncovering geometric mismatches between brains and ANNs, we argue that future models could benefit by aiming to preserve and increase intrinsic dimensionality throughout their hierarchy. Incorporating dimensionality-aware objectives into model training may be key to better approximate brain-like flexibility and semantic richness, and thus offer a principled benchmark for the next generation of biologically inspired artificial systems.

## 5   Limitations

One limitation of this study is our use of anatomical distance between region centroids as a proxy for hierarchical processing depth. Anatomical distance does not necessarily reflect functional or structural connectivity, particularly given the recurrent, non-feedforward organization of cortex. Future work could incorporate tractography or connectivity-based metrics to better capture hierarchical relationships, especially between non-adjacent regions. In addition, the BOLD signal can be influenced by anatomical factors such as sulcal depth [46], which may bias dimensionality estimates. Another limitation is that the ANNs we analyze are primarily feedforward and static, lacking the recurrent and temporally dynamic computations characteristic of biological vision; this could be addressed by evaluating recurrent architectures trained on temporal stimuli, such as video. Finally, early visual regions in the brain (e.g. V1–V3) are strongly structured by topographic features such as eccentricity and polar angle, which are not explicitly modeled in our approach. Future work could address this by testing topographic ANNs that better reflect the spatial organization of early visual cortex [53].

## 6 Acknowledgments

We thank Bradley Love, Heida Sigurðardóttir, Paolo Muratore, Gemma Roig and all members of the ARENA unit for helpful discussions. This research was supported by the German Research Foundation (DFG) - DFG Research Unit FOR 5368 ARENA. MW received funding from the Connected Minds Postdoctoral Fellowship (supported by CFREF), and the Deutsche Forschungsgemeinschaft (DFG, German Research Foundation) – 414985841.

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

# Supplementary Material

## A1    Compute resources and code availability

All code and scripts necessary to reproduce the experiments and figures are publicly available at `https://github.com/sgalella/dimensionality-brains-models`.

The code is designed to run on a local multicore machine (e.g., an 8-core setup) and supports parallel computing to accelerate processing. While running the analysis on a computing cluster can significantly reduce runtime, most of analyses can still be completed within a few hours on a standard local machine.

## A2    Additional fMRI data information

We replicated the dimensionality results using the BOLD5000 dataset, which contains fMRI data of four different subjects while viewing natural images of objects (from ImageNet-1K and COCO datasets) and scenes (from Scenes dataset) [7]. The total number of images observed by the subjects were 5234, from which 4916 were unique (for the fourth subject, the total number of images was 3108). We refer to the subjects using the same identifiers used in the original publication: CSI1, CSI2, CSI3, CSI4. We follow the same preprocessing details as for the THINGS fMRI dataset detailed in the main text. In total, we analyze the fMRI data of seven participants in two different datasets (see table A2).

| Name | Dataset | Sex | Age | Handedness |
|------|---------|-----|-----|------------|
| S1 | THINGS | Female | 29 | Right |
| S2 | THINGS | Male | 24 | Right |
| S3 | THINGS | Female | 23 | Right |
| CSI1 | BOLD5000 | Male | 27 | Right |
| CSI2 | BOLD5000 | Female | 26 | Right |
| CSI3 | BOLD5000 | Female | 24 | Right |
| CSI4 | BOLD5000 | Female | 25 | Right |

Table A2: THINGS and BOLD5000 subject information.

In the THINGS fMRI experiment, images were cropped to square format and varied in size (average: 996×996 pixels; minimum: 480×480 pixels; <1.8% smaller than 500 pixels). During presentation to human participants, images were shown on a gray background at a maximum visual angle of 10 degrees, with a 0.5-degree fixation crosshair overlaid [34]. Images were shown for 0.5 seconds followed by 4 seconds of fixation time. In BOLD5000, images were cropped to a size of 375 × 375 pixels, with a maximum visual angle of 4.6 degrees, and were shown for 1 second, followed by 9 seconds of fixation time [7].

### A2.1    Searchlight approach

To estimate local representational dimensionality in fMRI data, we employed a searchlight analysis [42, 9]. For each vertex (cortical surface space) or voxel (volume space), we defined a local neighborhood by selecting a fixed number of spatially contiguous vertices within a specified radius. We then computed ED and ID over the multivariate response patterns within each local patch, yielding a vertex-wise or voxel-wise map of local geometry. This approach preserves spatial continuity and captures topographic structure, such as retinotopy in early visual areas.

To assess the influence of spatial correlations on dimensionality estimates, we conducted control analyses using randomly sampled (non-contiguous) subsets of vertices within each region. While random sampling yielded higher absolute dimensionality values, likely due to reduced spatial autocorrelation, the relative hierarchy of dimensionality across regions remained intact, validating the robustness of our findings.

Here we parcellate the surface-based signal using the HCP atlas [see Fig. A1 30]. These regions can be broadly subdivided into primary, early, ventral, dorsal, and lateral areas (see Table A4).

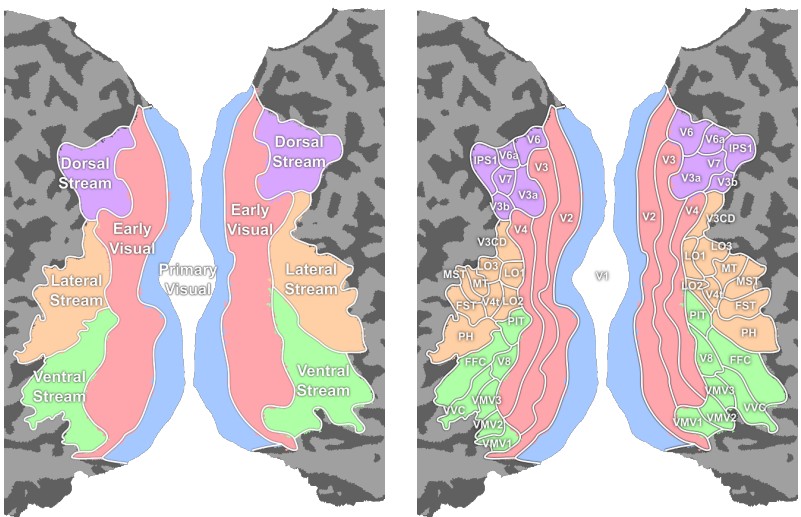

Figure A1: **Visual cortex parcellations from the HCP atlas [30]**

## A3 Additional artificial neural networks information

We include 36 different ANN models in our analysis (Table A5). Most models were trained on ImageNet-1K or Ecoset to perform object classification (see Table A5 Pretraining Dataset). While ImageNet-1K is one of the most popular dataset to train models for object classification [19], it includes many technical subclasses that do not represent how humans would classify objects (i.e., many exact dog breeds, instead of simply the class dog). Ecoset, by contrast, is curated to reflect categories that are more ecologically relevant to humans, with a focus on naturalistic, commonly encountered objects and animals, aiming to provide a more human-like visual learning environment for training neural networks [54]. Hence, comparing models trained with both datasets allows for a comparison, how human-centric categorization has an effect on dimensionality. We use the thingsvision implementation of Ecoset-trained models [56].

SimCLR was trained using a contrastive self-supervised learning objective on ImageNet-1K, where the model learns to bring augmented views of the same image closer in the representation space while pushing apart views from different images [8]. Finally, CLIP was trained using a contrastive image–text alignment objective on a large dataset of image–caption pairs, learning to associate images with their corresponding textual descriptions [59]. Additionally, we analyzed autoencoders (AE) trained on ImageNet-1K, Caltech256 [31], and Objects365 [64] to test how the dimensionality changes along the processing hierarchy for models with an unsupervised training regime. Moreover, we included biologically inspired models (CORnets [45]) to our analyses. These models were explicitly designed to mirror the hierarchical organization of the ventral visual stream, providing a direct computational analogue to cortical processing in the primate brain. Finally, we included a model trained on egocentric video data (EgoVLP; [48]) to capture a mode of visual experience more closely aligned with how humans perceive the world, emphasizing naturalistic, first-person perspectives.

We additionally include a ResNet-50 model trained to be robust against adversarial examples [22], referred to as ResNet-50 (Adversarial), and ResNet-50 models trained using data augmentation to be unbiased to textures [28], referred to as ResNet-50 (Fine-tuned on IN), ResNet-50 (SIN) and ResNet-50 (SIN + IN).

| Region ID | Region Name | Area |
|-----------|-------------|------|
| V1 | Primary Visual Cortex | Primary Visual |
| V2 | Second Visual Area | Early Visual |
| V3 | Third Visual Area | Early Visual |
| V4 | Fourth Visual Area | Early Visual |
| V8 | Eighth Visual Area | Ventral Stream |
| FFC | Fusiform Face Complex | Ventral Stream |
| PIT | Posterior Infero-Temporal Complex | Ventral Stream |
| VMV1 | Ventro-Medial Visual Area 1 | Ventral Stream |
| VMV3 | Ventro-Medial Visual Area 3 | Ventral Stream |
| VMV2 | Ventro-Medial Visual Area 2 | Ventral Stream |
| VVC | Ventral Visual Complex | Ventral Stream |
| V6 | Sixth Visual Area | Dorsal Stream |
| V3A | Area V3A | Dorsal Stream |
| V7 | Seventh Visual Area | Dorsal Stream |
| IPS1 | Intra-Parietal Sulcus Area 1 | Dorsal Stream |
| V3B | Area V3B | Dorsal Stream |
| V6A | Area V6A | Dorsal Stream |
| MST | Medial Superior Temporal Area | Lateral Stream |
| LO1 | Area Lateral Occipital 1 | Lateral Stream |
| LO2 | Area Lateral Occipital 2 | Lateral Stream |
| MT | Middle Temporal Area | Lateral Stream |
| PH | Parahippocampal Area | Lateral Stream |
| V4t | Area V4t | Lateral Stream |
| FST | Fundus of the Superior Temporal Sulcus | Lateral Stream |
| V3CD | Area V3CD | Lateral Stream |
| LO3 | Area Lateral Occipital 3 | Lateral Stream |

Table A4: Visual cortex regions and areas

For ViTs, we use either i) only the CLS tokens, or ii) the average of the rest of the tokens which we also refer to as GAP [20].

## A4 Additional fMRI results

### A4.1 Across subject comparison

While we mainly report detailed results for subject 1 of the THINGS database in the main text, we here also show the results for the other two THINGS subjects, as well as all subjects in the BOLD5000 (Figs. A2, A3). Across all, we observe very similar trends in increases in dimensionality along the visual processing hierarchy of the ventral visual cortex.

### A4.2 Dimensionality in the left and right hemisphere

Since the two hemispheres contain different numbers of vertices, we compute the correlation of sampled dimensionality across regions (see Fig. 2b) by averaging over 100 randomly selected vertices per region, repeated 100 times. We observe that the hemispheres in the visual cortex are quite similar, they process slightly different parts of the visual field. We therefore here confirmed that across all subjects and datasets, the dimensionality measures are similar within the left and the right hemisphere (Figs. A4, A5).

### A4.3 Effective vs. Intrinsic dimensionality in the brain

We observe highly correlated values for the linear (effective) and nonlinear (intrinsic) dimensionality estimation in the visual cortex for both datasets and subjects (Figs. A6, A7). This is different to

| Model | Architecture | Modality | Training Type | Pretraining Dataset | Objective function |
|---|---|---|---|---|---|
| AlexNet [44, 54] | CNN | Image | Supervised | ImageNet-1K Ecoset | Obj. Class |
| Inception V3 [71, 54] | CNN | Image | Supervised | ImageNet-1K Ecoset | Obj. Class |
| VGG 16 [65, 54] | CNN | Image | Supervised | ImageNet-1K Ecoset | Obj. Class |
| ResNet-18 [33] | CNN | Image | Supervised | ImageNet-1K | Obj. Class |
| ResNet-34 [33] | CNN | Image | Supervised | ImageNet-1K | Obj. Class |
| ResNet-50 [33, 28, 22, 54] | CNN | Image | Supervised | ImageNet-1K Ecoset | Obj. Class |
| ResNet-101 [33] | CNN | Image | Supervised | ImageNet-1K | Obj. Class |
| ResNet-152 [33] | CNN | Image | Supervised | ImageNet-1K | Obj. Class |
| SimCLR (ResNet-50) [8] | CNN | Image | Self-Supervised | ImageNet-1K | Contrastive |
| ViT-B/16 [20] | ViT | Image | Supervised | ImageNet-1K | Obj. Class |
| ViT-L/16 [20] | ViT | Image | Supervised | ImageNet-1K | Obj. Class |
| Swin-S [49] | ViT | Image | Supervised | ImageNet-1K | Obj. Class |
| Swin-B [49] | ViT | Image | Supervised | ImageNet-1K | Obj. Class |
| CLIP (ResNet-50) [59] | CNN | Image + Text | Self-Supervised | 400M Image-Text Pairs | Contrastive |
| CLIP (ViT-B/16) [59] | ViT | Image + Text | Self-Supervised | 400M Image-Text Pairs | Contrastive |
| CLIP (ViT-L/16) [59] | ViT | Image + Text | Self-Supervised | 400M Image-Text Pairs | Contrastive |
| AE (VGG16) [37] | CNN | Image | Unsupervised | Caltech256 ImageNet-1K Objects365 | Reconstruction |
| CORnet-Z [45] | CNN | Image | Supervised | ImageNet-1K | Obj. Class |
| CORnet-S [45] | CNN (Recurrent) | Image | Supervised | ImageNet-1K | Obj. Class |
| CORnet-RT [45] | CNN (Recurrent) | Image | Supervised | ImageNet-1K | Obj. Class |
| EgoVLP [48] | ViT | Video + Text | Self-Supervised | Ego4D | Contrastive |

Table A5: Models information.

the ANNs, where for the layers with highest effective dimensionality (later layers), the intrinsic dimensionality dropped (cf. Fig. 2).

### A4.4 Dimensionality increase in the ventral stream across all subjects

We further quantified the increase in dimensionality along the processing hierarchy in the ventral stream from primary, to early, to the ventral areas. We quantified the hierarchy using the average geodesic distance from the center of the primary visual cortex. Generally, we find an increase along the hierarchy for all THINGS and BOLD5000 subjects (Fig. A8).

### A4.5 Replication of the representation of low- and high-level features in the brain

We repeated all prediction analyses, including stimuli, low-level image features, categories, and abstract concepts, for the other two subjects in the THINGS database. For category decoding, we selected 100 of the 720 categories observed by each subject. Each category was represented by 12 different images, resulting in a total of 1,200 images. For stimulus decoding, we used a subset of 100 images, each shown 12 times across sessions, also totaling 1,200 image presentations. For decoding categories we use logistic regression, and for decoding stimuli, components and concepts we use ridge regression. All decoding analyses were performed using leave-one-session-out cross-validation.

We again find a reduced representation of stimuli and low-level features in later brain areas of the ventral stream (Figs. A9a, A10a, cf. Fig. 3a in the main text). The decoding performance of the categories and abstract concepts increases with increasing distance to the primary visual cortex (Figs. A9b, A10b, cf. Fig. 4a in the main text).

## A5 Additional ANN results

In the main text we only show the changes in ED and ID along the hierarchy for two example ANNs (ResNet50, ViT). We include the detailed ED and ID results (Figs. A11, A12), as well as for the categorization performance (Fig. A13).

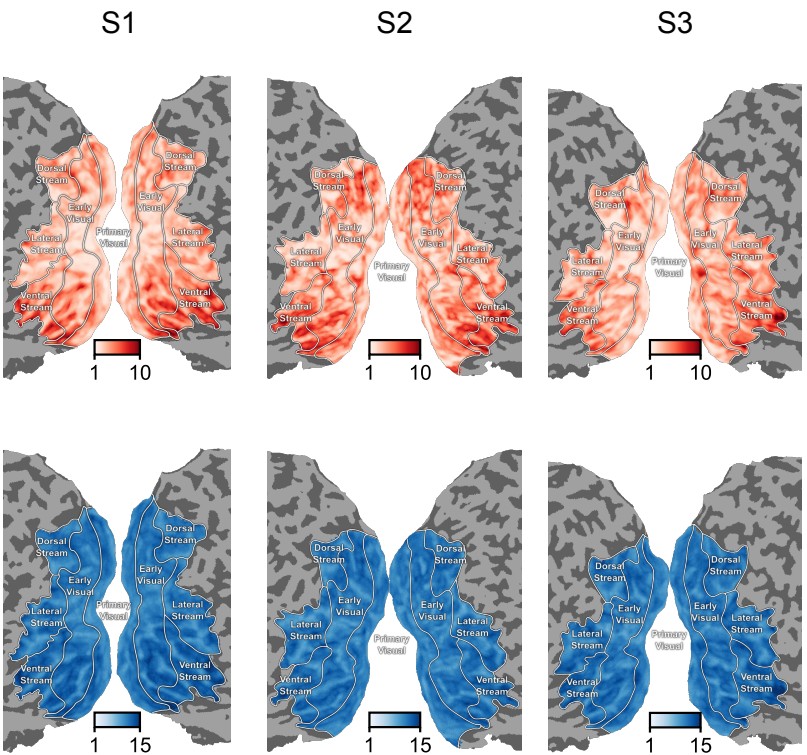

Figure A2: **Effective and intrinsic dimensionality for all three subjects of the THINGS database using 50 vertices.**

## A6 Robustness analyses in fMRI data

### A6.1 Varying radii for the searchlight approach

To test the robustness of the dimensionality computations to different radii of the searchlight, we report the results with three different radii (25, 50, 100 vertices (see Fig. A14) that approximately translate to 5, 7, and 10 mm of cortex (Table A7). As expected, we find that with smaller radii, the dimensionality is lower (Fig. A15), due to the reduced amount of possible dimensions. However, in both, effective and intrinsic dimensionality, we observe the same increasing trend towards the ventral regions. Again, effective dimensionality varies more than the intrinsic dimensionality.

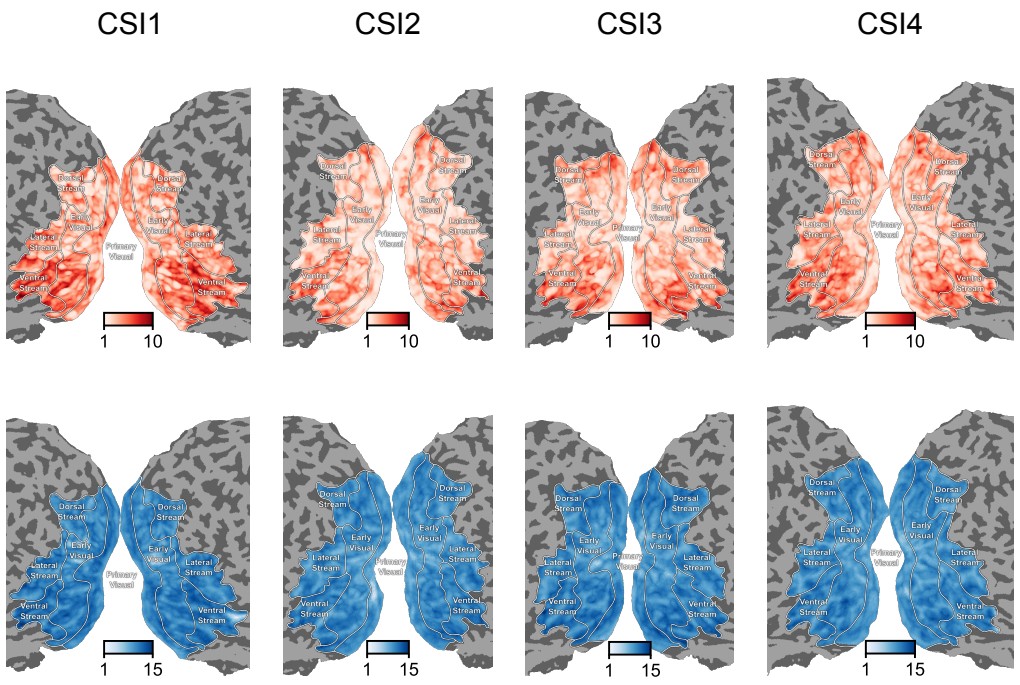

Figure A3: **Effective and intrinsic dimensionality for all three subjects of the BOLD5000 database using 50 vertices.**

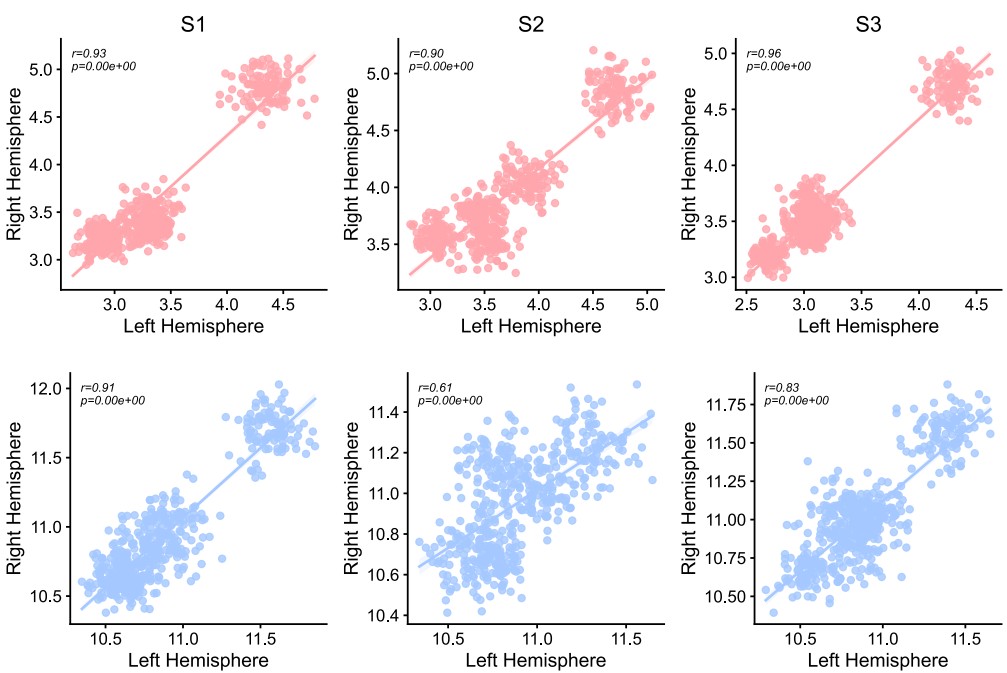

Figure A4: **Correlation between the dimensionality of the left and right hemisphere across brain regions for all THINGS subjects. Top: effective dimensionality, bottom: intrinsic dimensionality.**

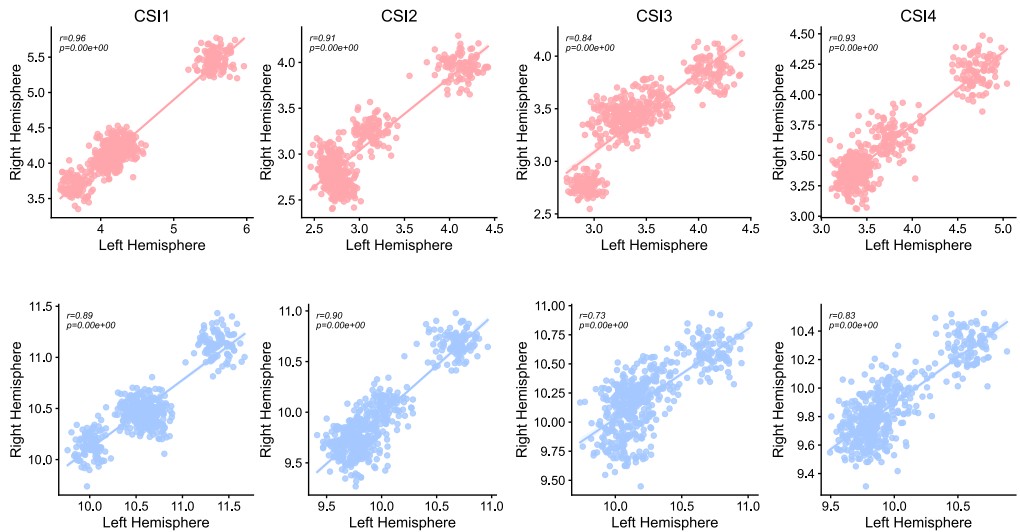

Figure A5: **Correlation between the dimensionality of the left and right hemisphere across brain regions for all BOLD5000 subjects. Top: effective dimensionality, bottom: intrinsic dimensionality.**

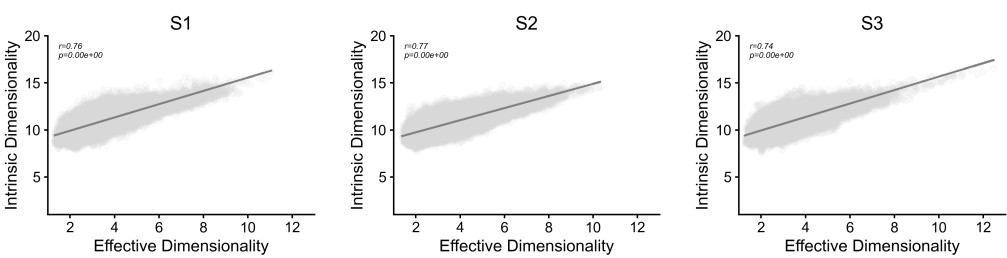

Figure A6: **Correlation between effective and intrinsic dimensionality of vertices for THINGS subjects.**

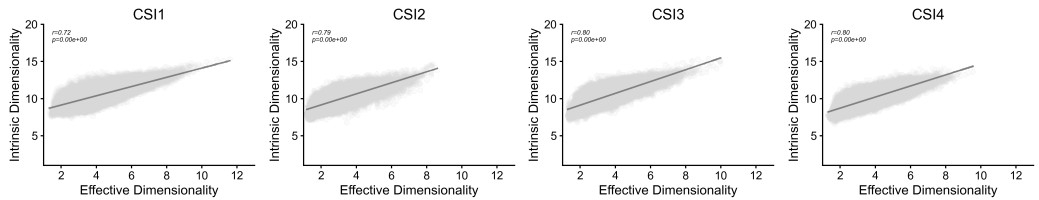

Figure A7: **Correlation between effective and intrinsic dimensionality of vertices for BOLD5000 subjects.**

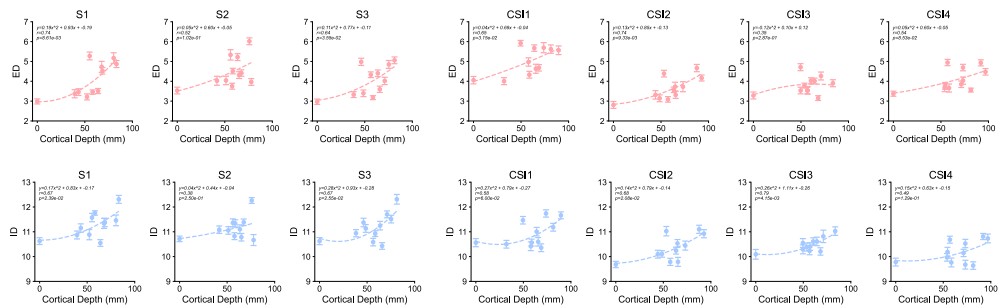

Figure A8: **Increase in effective (top) and intrinsic (bottom) dimensionality in all subjects of the THINGS and BOLD5000 databases**. Cortical distance is computed as the average geodesic distance between the center of V1 to the other areas. Each dot represents a area.

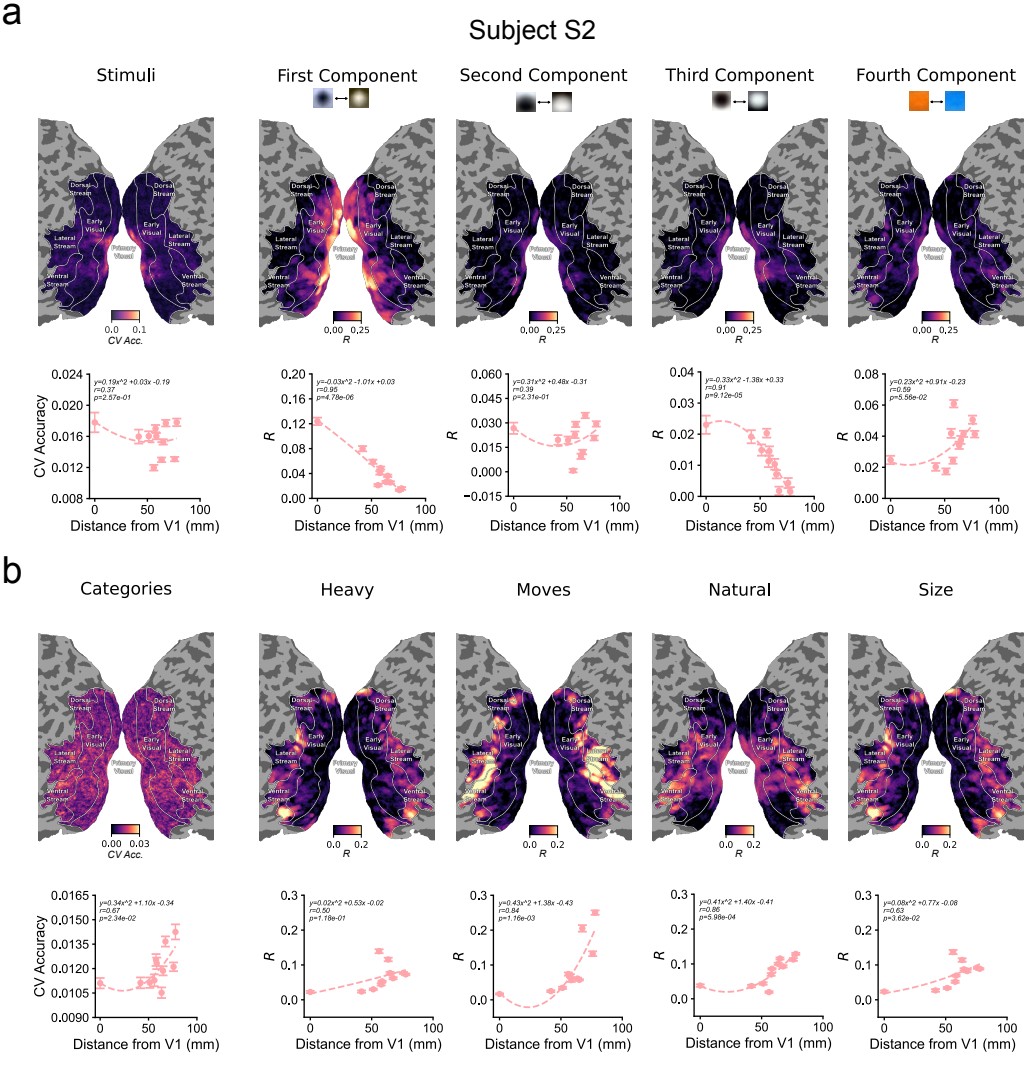

Figure A9: **Low-level (a) and high-level (b) feature decoding in THINGS Subject 2.**

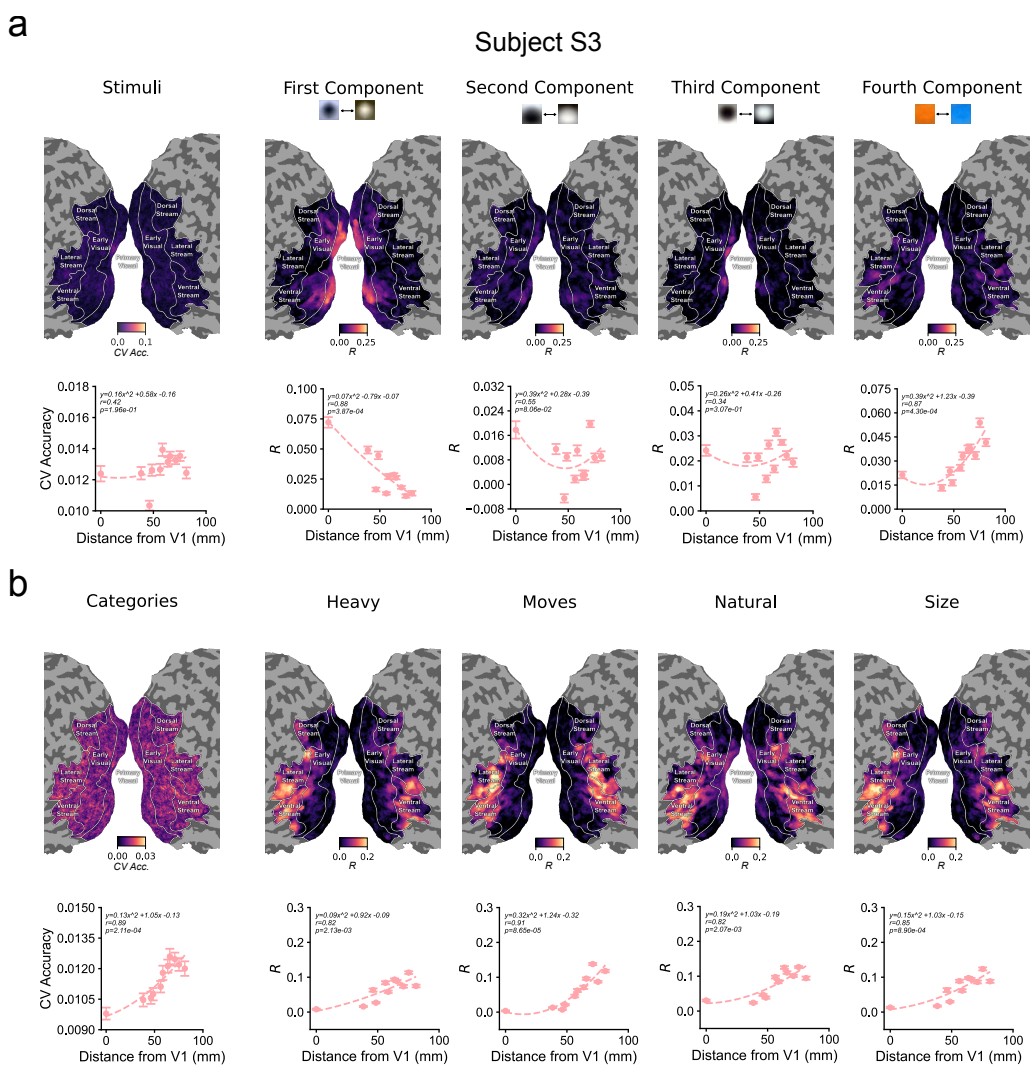

Figure A10: **Low-level (a) and high-level (b) feature decoding in THINGS Subject 3.**

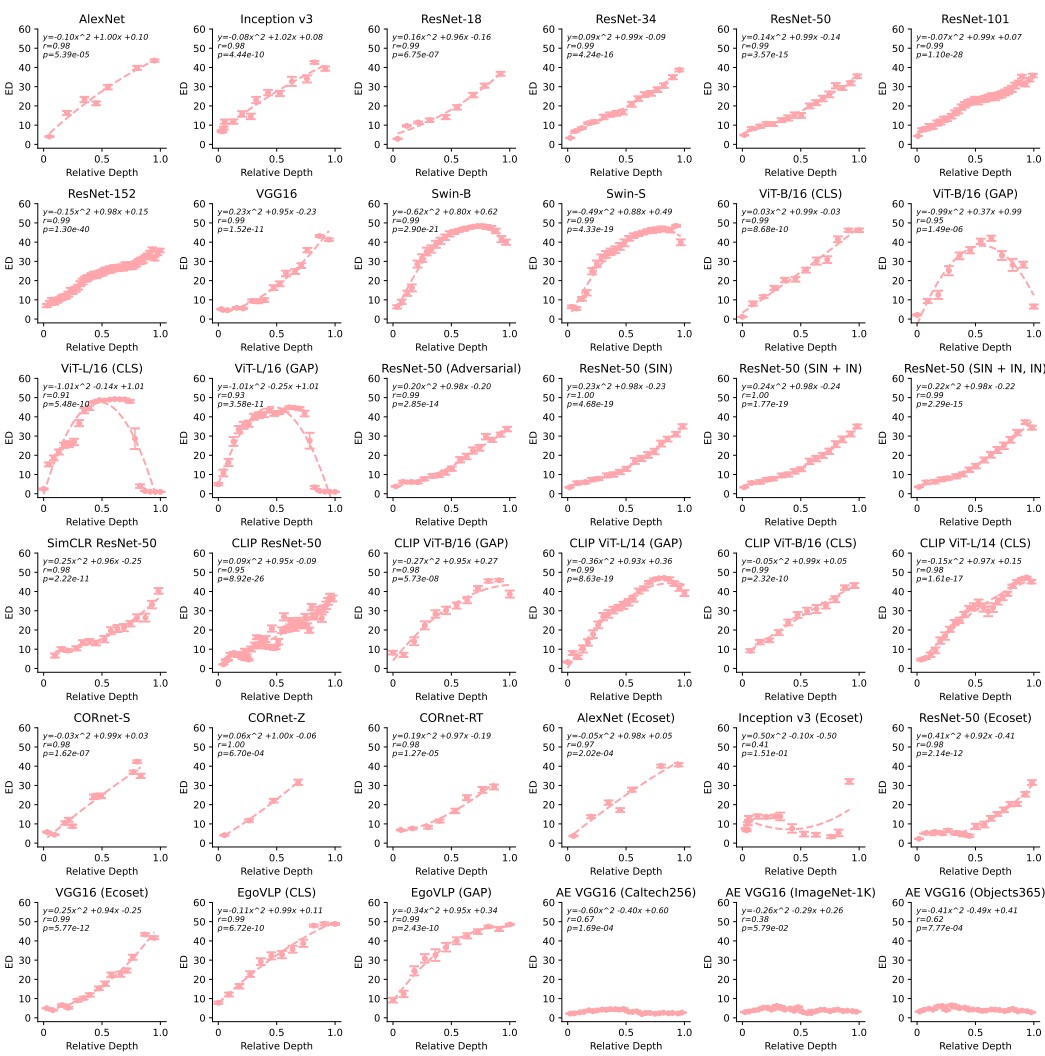

Figure A11: **Effective dimensionality changes along the hierarchy for all tested ANNs with global average pooled features.** Relative depth indicates the normalized index of the network layer. The error bars represent the standard deviation of the dimensionality calculated for 100 random subsets of 50 globally pooled features per layer. The models are ordered by: generic convolutional neural networks, ViTs, self-supervised models, biologically-inspired models, models with more human-like training data, and unsupervised autoencoders.

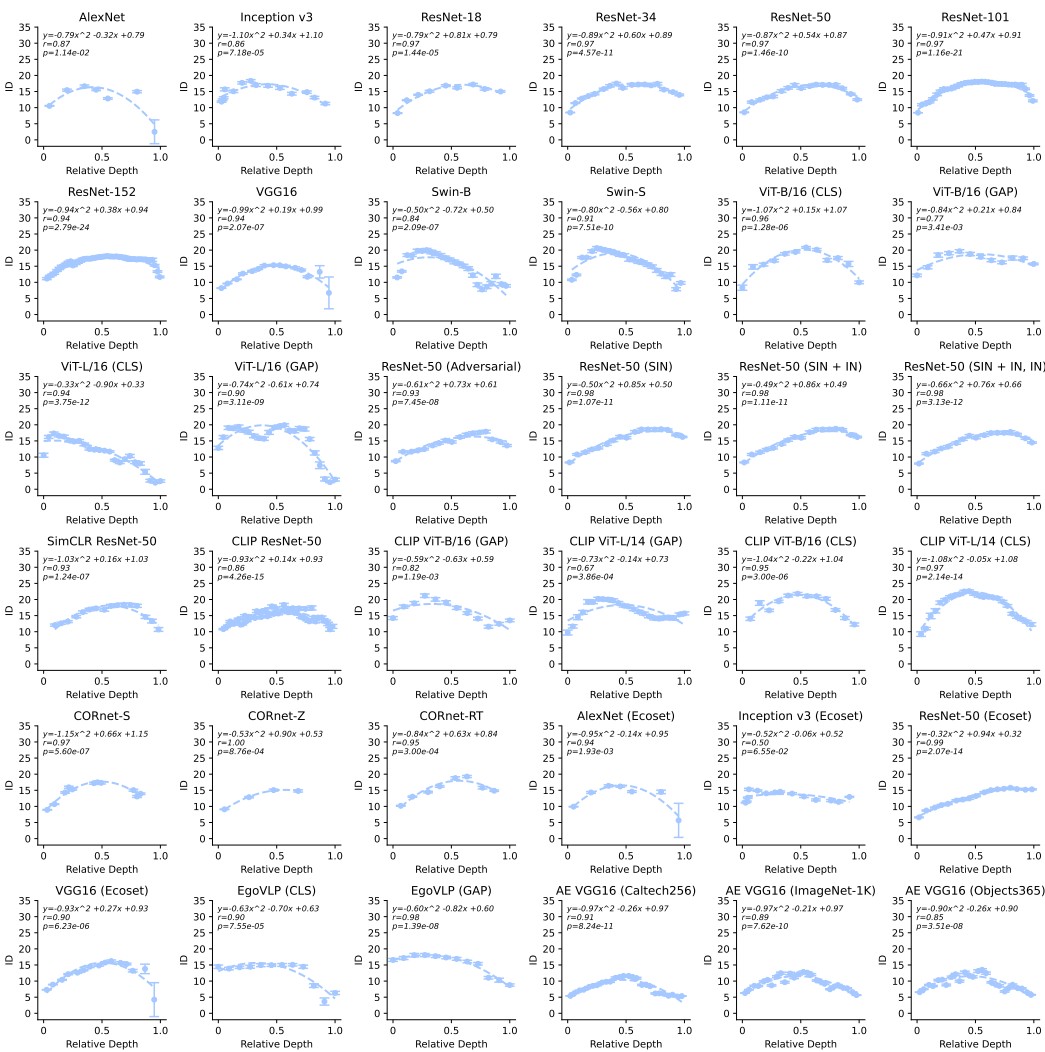

Figure A12: **Intrinsic dimensionality changes along the hierarchy for all tested ANNs with global average pooled features.** Relative depth indicates the normalized index of the network layer. The error bars represent the standard deviation of the dimensionality calculated for 100 random subsets of 50 globally pooled features per layer. The models are ordered as in Fig. A11.

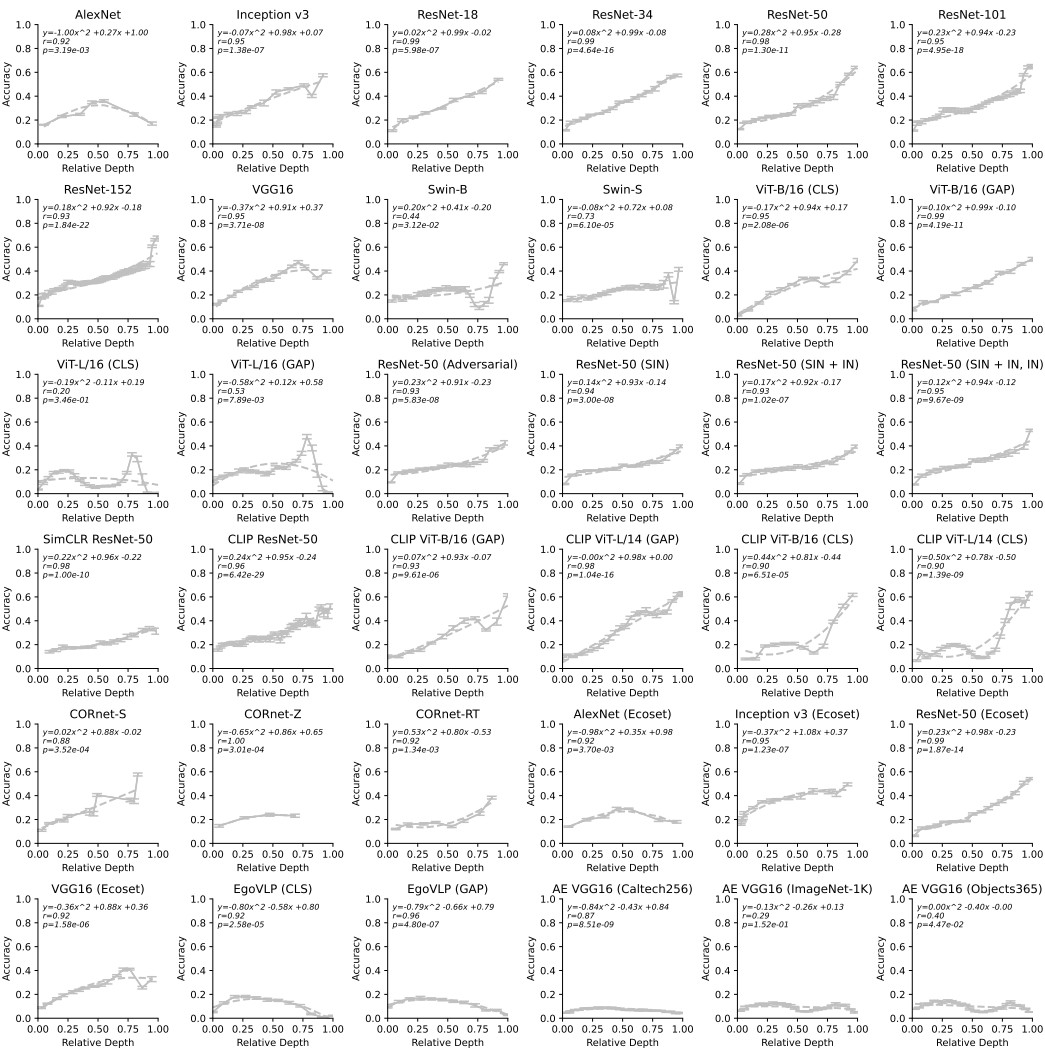

Figure A13: **Category decoding performance changes along the hierarchy for all tested ANNs with global average pooled features.** Relative depth indicates the normalized index of the network layer. The error bars represent the standard deviation of the performance calculated for 100 random subsets of 50 globally pooled features per layer. The models are ordered as in Fig. A11.

## 25 vertices    50 vertices    100 vertices

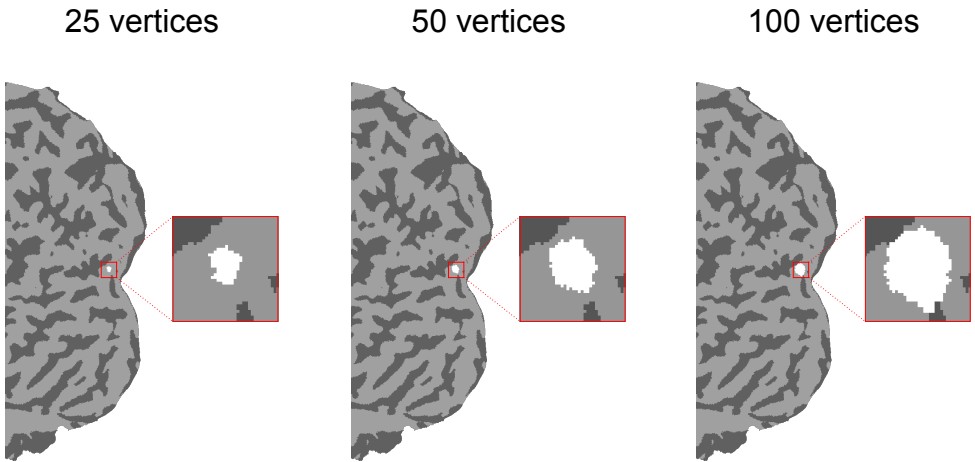

Figure A14: **Representation of the size of the searchlight on the cortical flatmap for Subject 1 from the THINGS fMRI dataset: 25, 50, and 100 vertices.**

## Subject S1

### 25 vertices    50 vertices    100 vertices

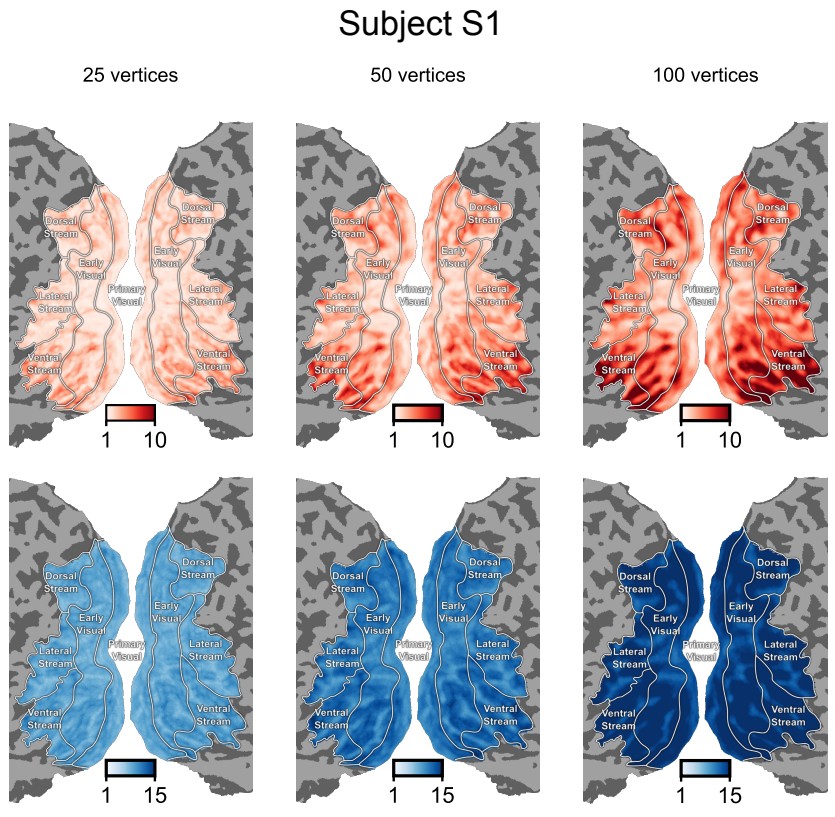

Figure A15: **Lower dimensionality with smaller neighborhood sizes**. For the neighborhood sizes shown in Fig. A14, we again compute the dimensionality across the whole visual cortex, and observe similar patterns as reported in the main text.

| Vertices | Left Hemisphere | Right Hemisphere |
|---|---|---|
| 25 | 4.97 ± 0.05 mm | 4.96 ± 0.05 mm |
| 50 | 7.06 ± 0.09 mm | 7.04 ± 0.08 mm |
| 100 | 10.01 ± 0.12 mm | 9.97 ± 0.12 mm |

Table A7: Average size (in mm) and standard deviation of the searchlight area using 25, 50, and 100 vertices.

## A6.2 Different cortical depth estimates

As the functional form of dimensionality increase can be affected by the type of cortical measure and order of the regions we additionally i) compute the Euclidean distance on the cortical surface and, ii) convert the geodesic distance into ranks instead of exact measures. We find Euclidean distance to be highly correlated with geodesic distance (Fig. A16a left). Similar to the results reported in the main text (Fig. 2). While the geodesic rank reveals a more linear increase in ED, ID becomes highly variable with an unclear structure (Fig. A16b).

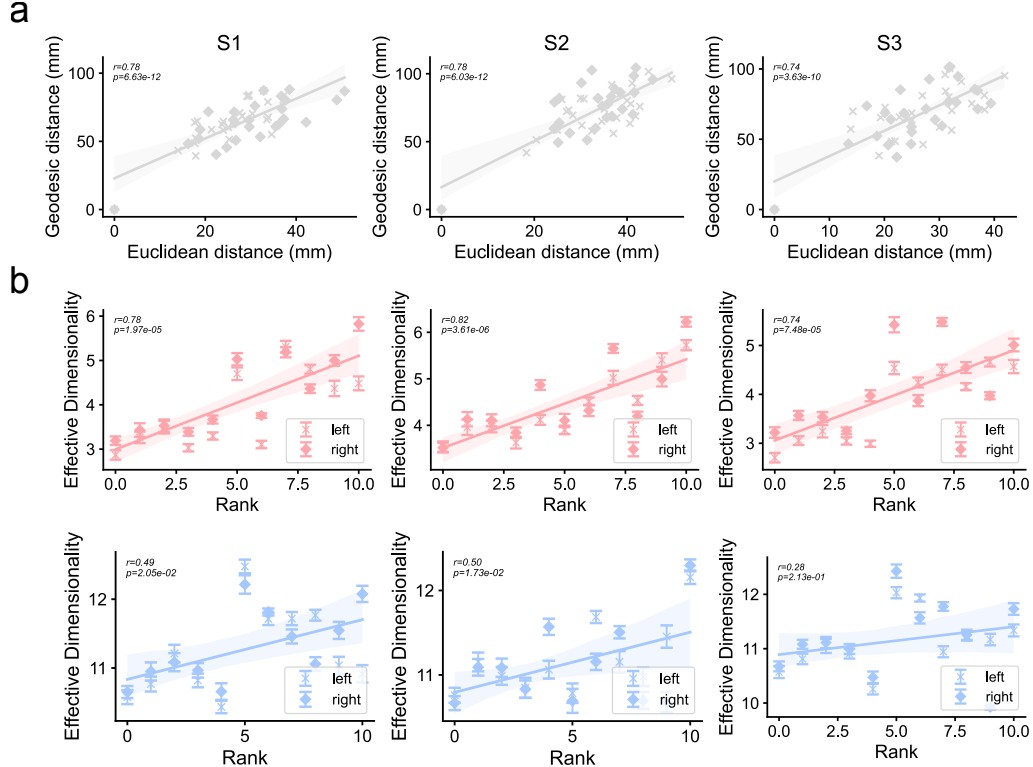

Figure A16: **Comparing different measures of cortical distance.** a) Left: High similarity between geodesic and Euclidean distance to compute cortical distance. b) Using the rank of geodesic distances instead of the exact distances also shows an increase along the hierarchy.

## A6.3 Random sampling in fMRI

In the main text, we report results using a searchlight approach to compute the dimensionality. Both, ED, ID, and decoding accuracy are generally higher in ANNs than in fMRI data using the searchlight approach, partly due to spatial correlations in fMRI vertices and noise reducing decoding accuracy. To test whether the differences in dimensionality are affected or even driven by the correlative structure

of the fMRI signal in nearby regions, we repeated the dimensionality computation sampling randomly vertex activity within an area (primary, early, ventral, lateral, dorsal). We find that even with this random sampling approach the ventral areas have a higher dimensionality than all other areas (Fig. A17), confirming our results with the searchlight approach and suggesting that the dimensionality is only slightly affected by the spatial correlative structure of the fMRI signal. Note, however, that random subsampling of vertices increases brain ED values, making them more comparable on an absolute scale to ANN results.

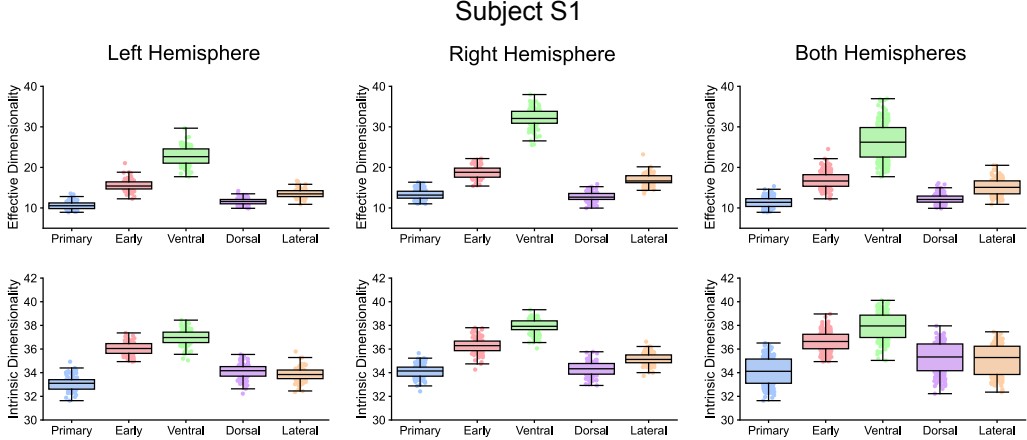

Figure A17: **Calculation of dimensionality sampling random vertices of the different areas instead of using neighbors of vertices. Top: effective dimensionality, bottom: intrinsic dimensionality.**

## A7 Robustness analyses in ANNs

### A7.1 Random sampling and random projections in ANNs

How features in an ANN are preprocessed when studying brain alignment can have a significant impact on the results [41]. In dimensionality analyses, it is important to restrict the number of features across layers or regions to ensure valid comparisons. A common approach is to use random projections, which preserve the pairwise distances in the data while reducing dimensionality. While this method retains overall geometric properties, it also mixes unit activations through arbitrary linear combinations, potentially distorting unit-level selectivity, which may be critical for interpreting brain–model correspondence [41].

Previous work has linked individual ANN units to single neurons in the brain [78], supporting unit-level comparisons. However, the fMRI signal reflects a population-level average across thousands of neurons within a voxel, suggesting a mismatch in spatial scale. To address this, we adopt global average pooling, which compresses feature maps into coarse-grained, image-level representations that potentially better approximate the aggregate nature of fMRI signals. This approach preserves semantic content while discarding spatial specificity, similarly to the spatial integration performed by millimeter-scale cortical sampling in fMRI. To evaluate how preprocessing affects dimensionality estimates, we computed both effective and intrinsic dimensionality using three approaches: (i) global average pooling with feature map subsampling, (ii) randomly sampled units that preserve selectivities (unit-level features), and (iii) random projections that preserve geometric structure while reducing feature dimensionality for three example models (ResNet50, SimCLR, CLIP). To compare the methods, we subsample images from the dataset to reduce computational costs. We sampled 500 images from the THINGS dataset, selecting only those observed by the subjects, and repeated each analysis 100 times per layer. For subsampling, we selected 50 pooled feature maps or units, while for random projections, we reduced the dimensionality to 50. Note that due the smaller number of images we used, the curves obtained here for global average pooling differ from those in Figure 2.

For effective dimensionality, we find that only global average pooling produces the increase across layers that mirrors the pattern observed in the ventral visual stream of the brain (Fig. A18). In contrast, unit sampling and random projections both yield the previously reported expansion–compression trend, even for effective dimensionality. For intrinsic dimensionality, this expansion–compression pattern persists across all preprocessing methods (Fig. A19), suggesting that nonlinear representational geometry in ANNs is shaped more by task constraints and training objectives than by feature selection alone.

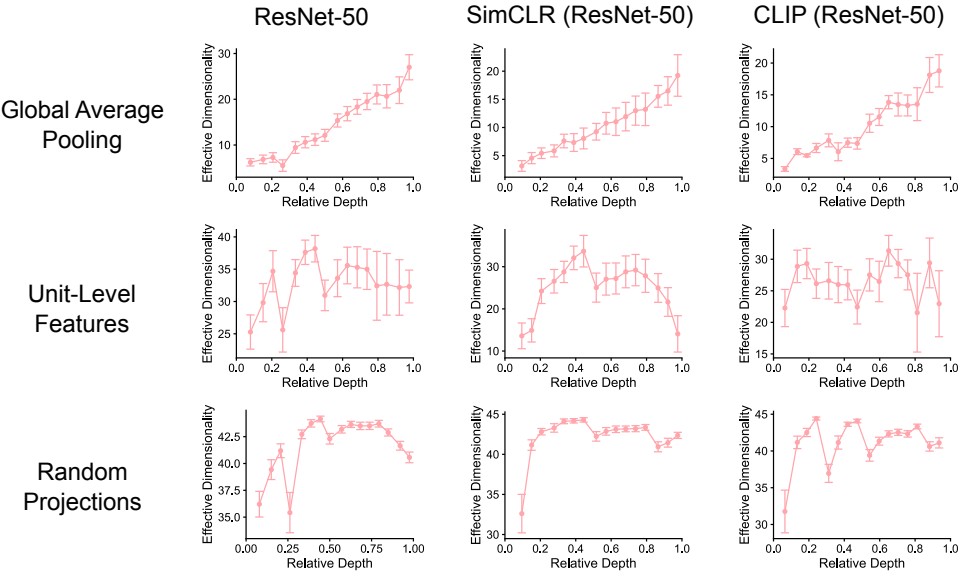

Figure A18: **Effective dimensionality in ANNs using different strategies to estimate dimensionality.** Note that differences to other reported results in Fig. A11 stem from a subsampling of only 500 images.

## A7.2 Identifying brain–model correspondences to contextualize dimensionality trends

Previous work hypothesized ANNs to be models of the ventral visual stream showing that ANNs share many similarities with biological brains, including hierarchical processing [e.g., 5, 78]. To test how dimensionality changes along the processing hierarchy in ANNs compared to the brain, we first performed two complementary analyses to identify which ANN layers align best with which cortical areas. First, we used BrainScore [63, 24, 52] to identify the layers that best correspond to V1, V2, V4, and IT in macaques (Fig. A20a top). We find that the decreasing trend in ID is already present in layers that best align with IT cortex (Fig. A20a bottom, red arrow indicates best IT-layer). Second, we leveraged our human fMRI data from the THINGS dataset to train linear encoding models to predict fMRI vertex responses from ANN features, going beyond the four BrainScore regions (Fig. A20b). Both approaches yielded similar results: early visual areas (e.g., V1–V4) are best predicted by early to mid layers, while higher ventral regions are best predicted by deeper layers. While these results were expected, the mapping between ANN depth and brain hierarchy allows us to make informed comparisons of dimensionality profiles across the two systems. These results further indicate that there is a fundamental mismatch in ID even when explicitly aligning the model layers to specific brain regions.

## A7.3 Effect of training on ANN dimensionality

To test whether the observed changes in dimensionality along the processing hierarchy of ANNs are an artifact of the underlying architecture, we also test untrained models (ResNets) with random initialization. For both, effective and intrinsic dimensionality, we find a dimensionality profile that strongly differs from the trained models. We observe a flat profile for ED along the hierarchy.

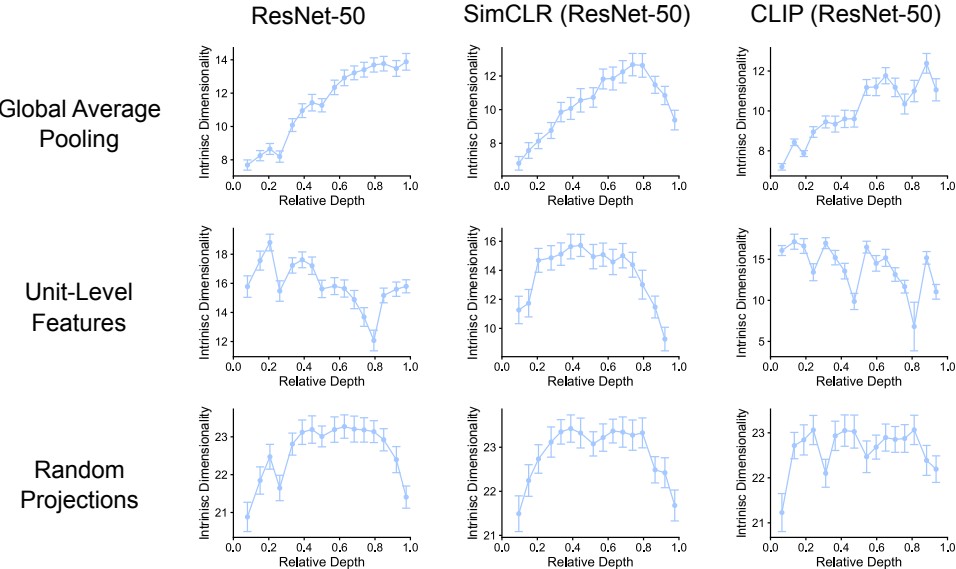

Figure A19: **Intrinsic dimensionality in ANNs using different strategies to estimate dimensionality.** Note that differences to other reported results in Fig. A12 stem from a subsampling of only 500 images.

However, ID increases in deeper layers when global average pooling is applied. This contrasts with prior studies without global average pooling, which reported a flat ID profile [e.g., 61], suggesting that pooling can enhance the expressivity of representations even in the absence of learned weights by aggregating weakly correlated features across the spatial map.

## A8 Robustness analyses of dimensionality computation

### A8.1 Effect of data preprocessing on dimensionality

We repeated the dimensionality analyses with different preprocessing strategies: i) with and without z-scoring (Fig. A21a), and ii) with and without mean-centering (Fig. A21b). Across both fMRI and ANN data, we found that dimensionality estimates were highly correlated. ID was particularly robust, while ED showed minor variation, likely due to the impact of mean-centering and z-scoring on the covariance matrix. In contrast, ID relies on relative distances that appear less sensitive to such transformations. These robustness checks support the stability of our main conclusions.

### A8.2 Effect of parameters on intrinsic dimensionality estimate

The the number of neighbors used to estimate the intrinsic dimensionality is an important hyper-parameter that could affect the results. We therefore repeated the analysis with different $k$-values ($k = 5, 10, 20$) and found that all results hold when changing the number of neighbors (Fig. A23).

## A9 Low-level feature representations

### A9.1 Extracting low-level feature representations in images

We performed PCA on the THINGS images viewed by subjects during the experiments to extract low-level visual features [73, 68]. Ordering the images by their loading on the first PCs reveals contrast, spatial frequency and color components (Fig. A24). The first four components already explain 43.67% of the total variance, highlighting the low-dimensionality of the input data. Due to

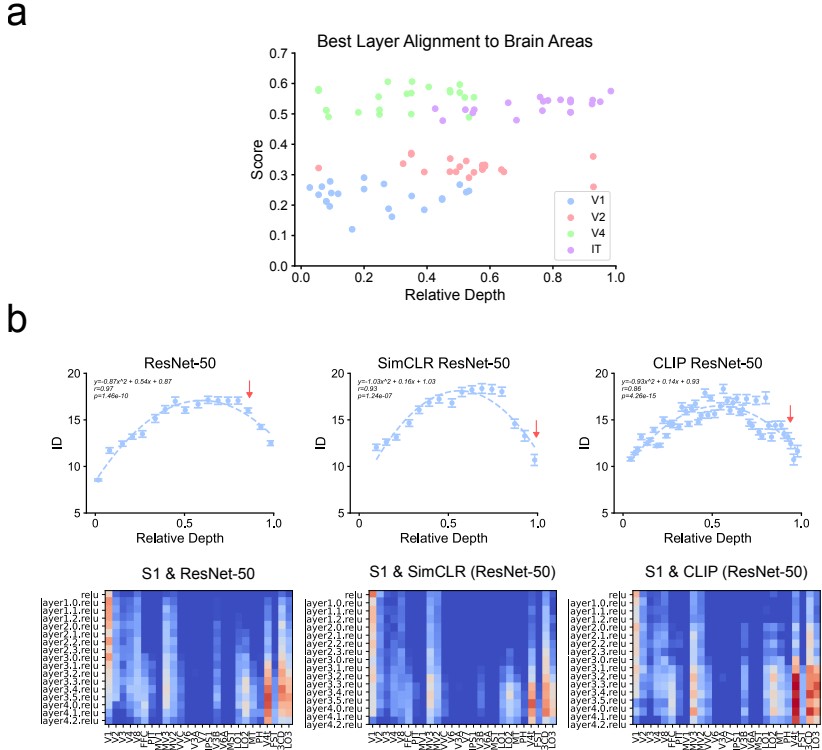

Figure A20: **Alignment between brains and ANNs**. a) We use BrainScore to find the layer that best aligns with V1, V2, V4, and IT cortex (top). Additionally we indicate the layer that most closely aligns with IT cortex (bottom, red arrow). b) Additionally, we train linear regression models to map each ANN layers' GAP activations to the fMRI features. Red regions indicate high alignment, while blue shows low alignment performance.

copyright restrictions, we do not display the original images. Instead, we project and show images from the THINGSplus dataset [69].

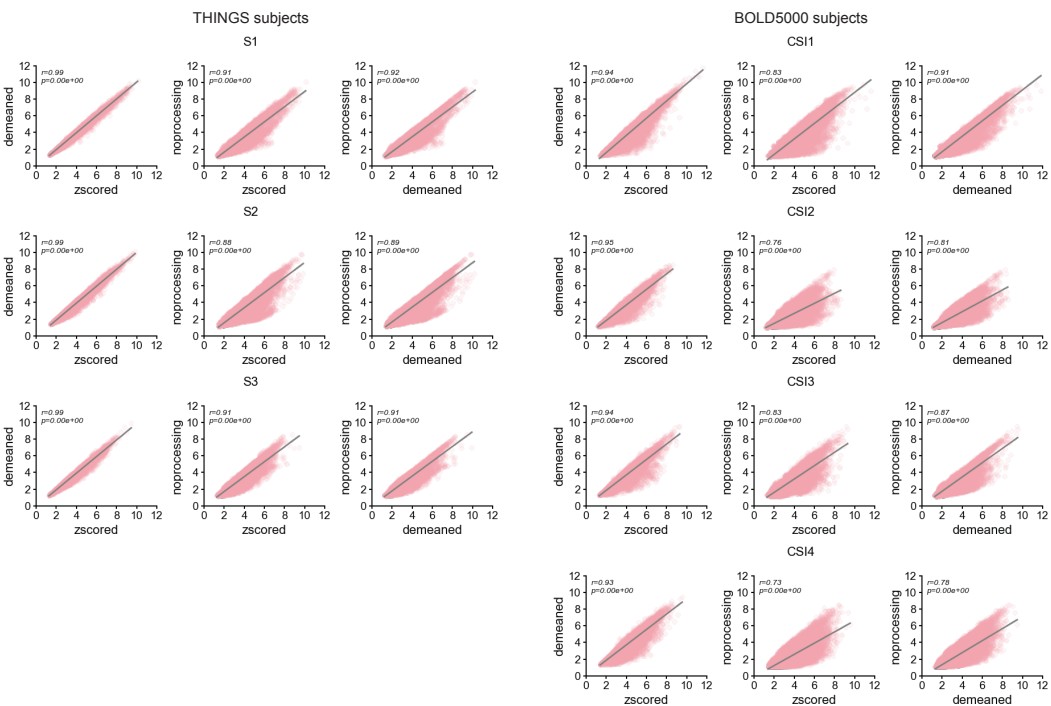

Figure A21: Effective dimensionality computations with and without z-scoring, and with and without mean-centering.

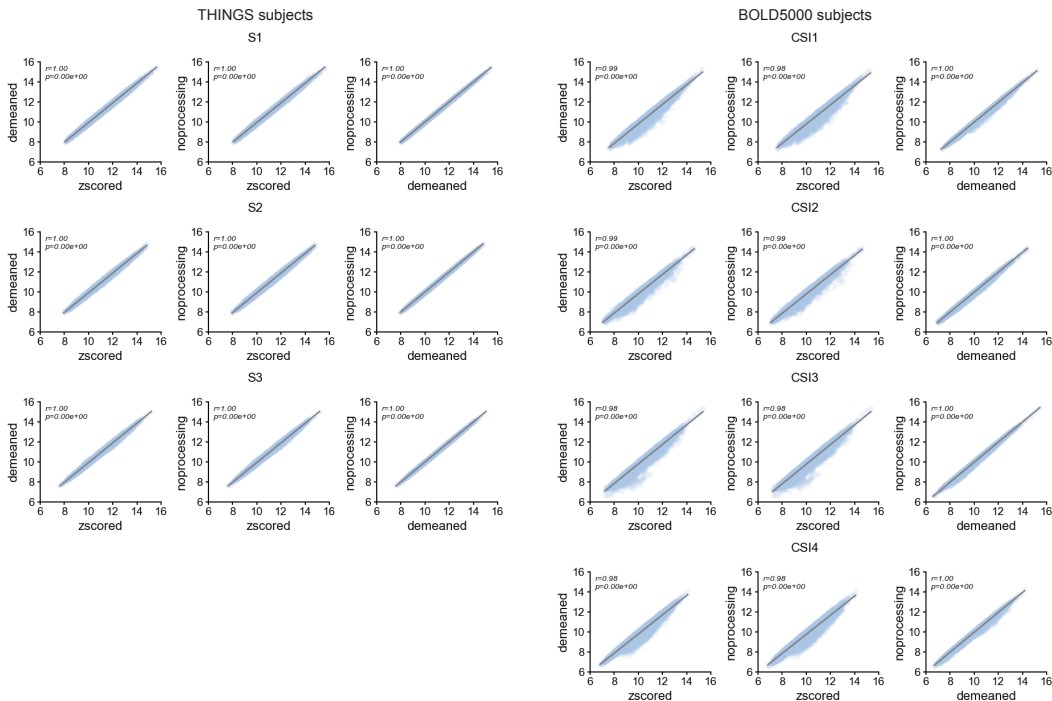

Figure A22: Intrinsic dimensionality computations with and without z-scoring, and with and without mean-centering.

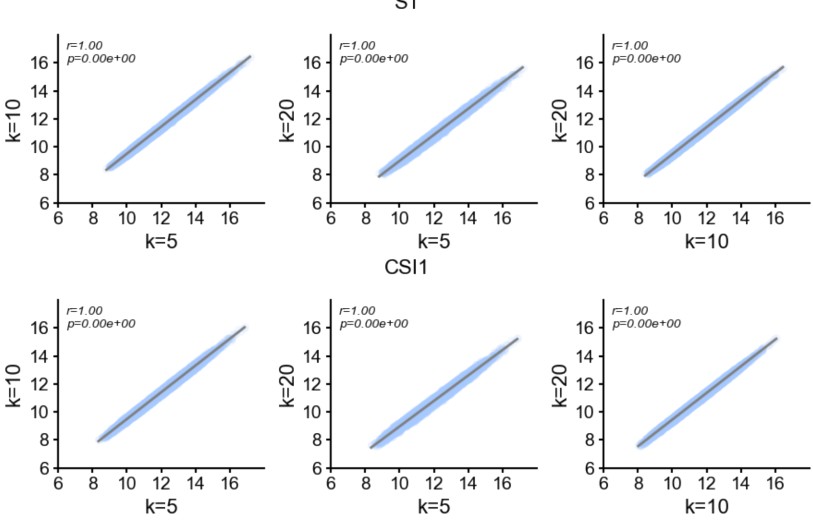

Figure A23: Different numbers of neighbors for the intrinsic dimensionality computation.

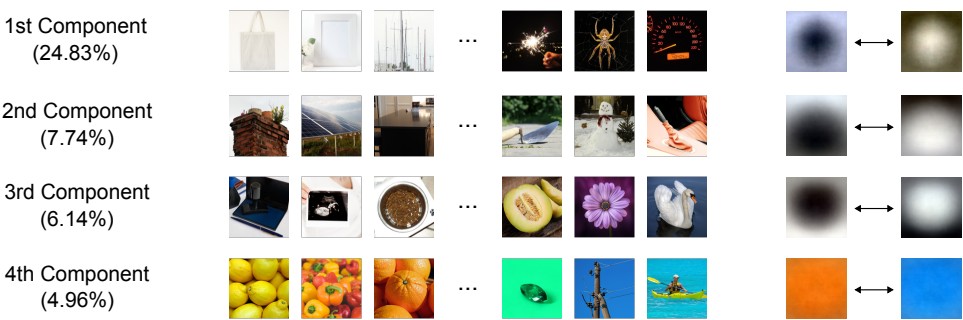

Figure A24: **Image-level PCA reveals low-level features in the images.**

