# OpenReview forum: "Dimensionality Mismatch Between Brains and Artificial Neural Networks"
_NeurIPS.cc/2025/Conference — NeurIPS 2025 poster_

### Official Review · Reviewer_WfsF · 2025-06-19

**Clarity:** 4
**Significance:** 2
**Originality:** 3
**Rating:** 6
**Confidence:** 4

**Summary:**

This study addresses the important question of how information is processed across hierarchical stages in both biological and artificial neural systems. The authors begin by clearly distinguishing between linear (effective) and non-linear (intrinsic) dimensionality. The work is focused on the vision modality. The authors compare different CNN-based networks with the different regions of the brain and compare their representational dimensionality. The results show that there is a striking mismatch between the ANNs and the human brain. In contrast to the brain where intrinsic dimensionality increases in a linear fashion, ANNs intrinsic dimensionality follows the canonical hunchback shape defined by an initial increase followed by a continuous decrease.

This is a very interesting and well-documented study aimed at understanding how the dimensionality of representations in the human visual cortex, specifically the ventral visual stream, evolves. Is it compression–expansion, continuous expansion similar to other species, or expansion–compression as observed in ANNs? Thus, the objective here is to investigate the commonalities and differences in representational geometry between biological and artificial systems.

Overall, the authors have presented a well-curated study with a primary emphasis on biological systems (neuroscience), while artificial systems receive comparatively limited attention and serve mainly as a secondary point of comparison. I will thoroughly explain this in the Strengths and Weaknesses section.

**Questions:**

1. What image resolution was used when testing Human subjects and ANNs? Was there a difference between the two? If yes, then why?
2. The authors define dimensionality later in the Methods section (Line 123), despite discussing it extensively earlier in the Introduction. Given that this is a computational neuroscience paper and is likely to attract readers from both machine learning and neuroscience backgrounds, it would benefit the reader if this definition were introduced earlier in the paper.
3. The authors provide no details about the pre-trained models. They should provide information about their objective functions and other necessary information for readers who are not familiar with the ANNs training objectives.
4. How did the authors calculate the relative depth of the ANNs as shown in the graphs?
5. The authors provide no information about the human subjects or their age groups. If the authors believ this information was included in the manuscript or the Appendix section, please refer to those line numbers in the comment.

**Ethical Concerns:**

["NO or VERY MINOR ethics concerns only"]

**Final Justification:**

The authors have performed additional experiments and generalized their findings to a different class of models, transformers. The new results are strong and therefore strengthen the original claims.

Furthermore, the authors have also addressed all the queries that were raised. Therefore, I am increasing my score.

**Limitations:**

The authors have not included any limitations in this paper. It is highly recommended to add a Limitation section.

**Paper Formatting Concerns:**

No Concerns

**Quality:**

3

**Strengths And Weaknesses:**

Strengths:

1. The paper conducts an extensive set of experiments on the biological side, specifically examining the human brain and its sub-regions in detail. The results are clean and well-presented for both biological and artificial systems.

2. The paper has a single objective, and it achieves that objective through the experiments presented in the main text. However, I have some concerns regarding the experiments conducted for the ANNs, which I will address below in the 'Weakness' section.

Weakness:
Due to the following major weaknesses, I have deducted two points. If the authors can provide follow-up experiments addressing some of these concerns, I will increase the score.

1. [MAJOR] Comparing ANNs with the human brain using pre-trained models may not be the most appropriate approach. While ANNs are typically trained on large datasets of static, isolated images (e.g., ImageNet), the human brain is exposed to continuous, smooth, and temporally structured visual input throughout life. This results in a significant mismatch in their respective visual diets (i.e., training data). As a result, it's unclear whether the differences reported in the study arise from this mismatch in training data or from deeper architectural or objective functions.

A potential solution is to train ANNs on egocentric datasets (such as first-person video) or in simulated environments that better mimic the visual diets of humans. While such datasets may not achieve perfect equivalence, they would allow for a closer and more meaningful comparison between biological and artificial systems.

2. [MAJOR] Does intrinsic dimensionality collapse in the deeper layers of all ANNs? The title and much of the language in the paper refer broadly to "artificial neural networks (ANNs)"; however, the experiments are conducted exclusively on convolutional neural networks (CNNs). While I have no objection to the exclusive use of CNNs for the current analyses, the framing may be misleading. As it stands, the paper appears to generalize its findings to all ANNs, including architectures such as Transformers, which have markedly different inductive biases and architectural features. Clarifying this distinction, or including additional experiments with non-CNN architectures, would strengthen the paper and ensure that its conclusions are correctly scoped.

Transformers have a fundamentally different architecture compared to CNNs. Unlike CNNs, Transformers do not rely on pooling layers, hierarchical structures, or hardcoded spatial biases. Recent studies have shown that Transformers can mimic key features of the brain. For example, they show representations similar to those of hippocampal cells, demonstrate neuron–astrocyte-like interactions, and are capable of learning object recognition when trained through the eyes of newborn animals.

Given these findings, it is important to include Transformers in the analyses presented in this paper. Specifically, it would be valuable to test whether the results reported here can be replicated using a Vision Transformer (ViT) model. It is acceptable for the ViTs to be pre-trained, just as the CNNs used in the current experiments were. Furthermore, it is also acceptable and valid if the ViTs produce different results; such differences would still offer meaningful insights into architectural and representational differences between models.
Without testing other ANN types, the claims in this paper offer a weak generalization.

3. [MINOR] Why is it that a randomly initialized ResNet-50 shows a similar ID pattern to that observed in Subject 1, while a trained ResNet-50 does not exhibit this pattern? I suggest repeating this experiment with different architecture sizes, such as ResNet-18, ResNet-34, and ResNet-101, just for the untrained case. It would be interesting to see whether the pattern observed in the untrained ResNet-50 is preserved across different architecture sizes.

---

> ### Author Rebuttal · Authors · 2025-07-30
>
> Thank you for your feedback and interest in our work. Your suggestions helped us broaden our analyses and strengthen our results. Please let us know if there is further clarification we can provide.
>
> **1. Models trained with a more human-like visual diet**
>
> Thank you for suggesting to include models trained with a more human-aligned visual diet. This was a highly valuable suggestion that substantially strengthened our study. In response, we extended our analyses with two additional experiments: i) models trained on a visual diet that better reflects human visual experience than ImageNet (Ecoset [1]), and ii) a model trained on ego-centric videos to capture object perception from a dynamic, first-person perspective (EgoVLP [2]). Interestingly, across both of these models, we observe effective (ED) and intrinsic (ID) dimensionality **both to increase across all layers**, without the late-stage ID compression found in ImageNet-trained models. Further, the Ecoset model only shows a weak drop in generalization performance to abstract features in the last layer. This suggests that training on more naturalistic visual statistics may encourage models to maintain richer, higher-dimensional representations throughout their processing hierarchy, supporting better generalization to abstract features and leading to geometric trends that more closely resemble those observed in the brain. The EgoVLP model, on the other hand, shows low, highly variable, and decreasing generalization to abstract features. It is important to note, however, that this model was trained on dynamic, egocentric videos that differ substantially from the static THINGS images used for evaluation. While EgoVLP may encode information in a high-dimensional subspace, the relevant feature separation might occur along dimensions that are not well captured by our test dataset. We are aware of datasets featuring human subjects watching naturalistic videos, as well as recent models specifically trained on such data. Incorporating these resources will allow for a more appropriate evaluation of egocentric and video-based models would be an interesting future direction. Overall, our new results highlight the importance of the visual diet in shaping representational geometry and suggest that aligning training data with human experience may be key to improving ANN–brain alignment.
>
> **2. Testing dimensionality collapse in non-CNN architectures**
>
> Thank you for your comment regarding the generalizability of our findings beyond CNNs. We agree that framing the results as representative of all ANNs was too broad given the focus on CNNs. To broaden the scope of our findings, we extended our experiments to include ViTs [3], Swin transformers [4], and CLIP-ViT [5]. We show that all models exhibit a similar pattern of ID collapse in their deeper layers. Moreover, the ED of ViTs and Swin transformers also generally drops in later layers, while CLIP-ViT shows an ED pattern that is more similar to the brain and CNN architectures with a steady increase in ED along the hierarchy. These findings suggest that the observed divergence between brain and ANN representational geometry is not exclusive to CNNs, but also applies to ViTs, reflecting a more general limitation of current feedforward vision architectures and, as shown above, visual statistics that deviate from the human visual diet. Note that in response to other reviewer feedback, we extended our analyses to also include both **biologically inspired (CORNet [6]), unsupervised models (Autoencoders [7]), and other CNN-architectures (AlexNet, Inception, VGG16, ResNets)**. The unsupervised models showed an increase in ID throughout the encoder, but a marked drop in the decoder, while ED remained relatively stable across all layers. This pattern indicates a different type of mismatch to brain data compared to supervised models. Across all CORNet models, ED consistently increased along the hierarchy. For ID, CORNet-Z exhibited a continuous increase, while the recurrent models show the typical drop in late layers. However, CORNet-Z is very shallow, which might confound the results. All other CNN architectures exhibited the same dimensionality patterns as ResNet50. Overall, these extensions highlight that the dimensionality dynamics vary substantially across architectures and objectives, yet key mismatches to the brain persist under most conditions. We will incorporate these results in the revised manuscript and adjusted the title and text to more accurately reflect the scope of our analyses. We also appreciate the additional contextualization and studies mentioned by the reviewer and will include these in the manuscript.
>
> **3. Different untrained models**
>
> Thank you for suggesting to test different model architectures to better understand the indeed surprising increase of ID in the untrained models. We observe a similar step-wise increase across all untrained ResNet models. While deeper architectures maintain a prolonged plateau in ID in the intermediate layers, all models consistently show a sharp increase in ID in the final two to three layers. Note however that across all untrained models, we observed markedly lower dimensionality compared to trained models, consistent with prior findings [8]. We will comment about this in the revised manuscript and will add the additional analyses to the supplement.
>
> **4. Image resolution**
>
> In the fMRI experiment, images from the THINGS database were cropped to square format and varied in size (average: 996×996 pixels; minimum: 480×480 pixels; <1.8% smaller than 500 pixels). During presentation to human participants, images were shown on a gray background at a maximum visual angle of 10 degrees, with a 0.5-degree fixation crosshair overlaid [9]. For the ANN analyses, we followed the preprocessing procedures used during model training: images were typically rescaled to 224×224 pixels for input into the models. Thus, there is indeed a resolution difference between the human and ANN input. This reflects standard practice in computational neuroscience, where model inputs are adjusted to match the conditions of model training, ensuring valid and meaningful feature extraction. Importantly, since both humans and models were exposed to the same images (despite different resolutions), and our focus lies in comparing representational trends across stages rather than precise pixel-level responses, we do not expect this difference to qualitatively impact the main conclusions. Also, our main results regarding differences between models and the brain relate to the later processing stages that are less affected by image resolution and visual degree as early regions. Nonetheless, we will clarify and comment on this in the text.
>
> **5. Late introduction of dimensionality**
>
> Thank you for the comment. We will introduce these concepts earlier in the text.
>
> **6. Details about pre-trained models**
>
> We agree that more information about the pre-trained models will be helpful for readers not familiar with ANNs and their training objectives and will add this in the revised version. In short, the ResNet50 was trained on ImageNet to perform object classification. SimCLR was trained using a contrastive self-supervised learning objective on ImageNet, where the model learns to bring augmented views of the same image closer in the representation space while pushing apart views from different images. Finally, CLIP was trained using a contrastive image–text alignment objective on a large dataset of image–caption pairs, learning to associate images with their corresponding textual descriptions.
>
> **7. Relative depth calculation**
>
> Thank you for pointing out the lack of detail regarding the computation of relative depth, a point that was also raised by other reviewers. First we extracted the activations of all ReLU layers (or GeLU in transformer based models). We then normalized the layer indices of these extracted layers by the total number of layers in the model. We will clarify this procedure in the Methods section of the revised manuscript.
>
> **8. Information about the human subjects**
>
> We did not include more details about the human subjects in the appendix but will correct this in the revised supplement. Overall, the THINGS fMRI study included three participants (2 female, 1 male) with a mean age at the beginning of the study of 25.33 years. The behavioral study to generate the perceptual scores for the abstract features (e.g., heavy, moves, etc.) included 4,156 participants (2,418 female, 1712 male, and 26 other) with a mean age of 36.02±11.80 years. The BOLD5000 fMRI data comprises four individuals (3 female, 1 male) with a mean age of 25.5 years.
>
> **9. Missing limitations section**
>
> We agree that including a dedicated limitations section will enhances the clarity and transparency of the paper, and will add one in the final version. While some limitations, e.g., the use of geodesic distance, were already addressed in the Discussion, we will make these points more explicit and consolidate them into a clearly labeled limitations section.
>
> **References**
>
> [1] Mehrer et al., 2021. An ecologically motivated image dataset for deep learning yields better models of human vision.
>
> [2] Lin et al., 2022, Egocentric Video-Language Pretraining.
>
> [3] Dosovitskiy et al., 2020. An image is worth 16x16 words: Transformers for image recognition at scale.
>
> [4] Liu et al., 2021. Swin transformer: Hierarchical vision transformer using shifted windows.
>
> [5] Radford et al., 2021. Learning transferable visual models from natural language supervision.
>
> [6] Kubilius et al., 2018. Cornet: Modeling the neural mechanisms of core object recognition.
>
> [7] Github: Horizon2333, imagenet-autoencoder.
>
> [8] Recanatesi et al., 2019. Dimensionality compression and expansion in deep neural networks.
>
> [9] Hebart et al., 2023. Things-data, a multimodal collection of large-scale datasets for investigating object representations in human brain and behavior.

---

> > ### Comment · Reviewer_WfsF · 2025-08-01
> > **Response to rebuttal**
> >
> > I thank the authors for addressing each one of my queries. My response to your points is below.
> >
> > **1. Running models on egocentric data:** It's great that the authors were able to evaluate new models that were trained on more realistic egocentric datasets. The authors suggest that aligning training data with human experience may be key to improving ANN–brain alignment. I strongly believe that this is a remarkable finding in bridging the gap between human and machine-level intelligence. I strongly advise the authors to add this additional experiment as a separate section in their manuscript and direct readers to these findings. I also agree that the test images significantly differ from the egocentric experiences and therefore ask the authors to add this in their future work to test such models on various other test sets. This new experiment deserves a point.
> >
> > **2. Running transformer and models suggested by other reviewers:** I appreciate that the authors considered adding non-CNN-based models and evaluated them. Given that the results generalize to ViTs is a great finding that aligns with many past research studies showing that transformers show similar computational patterns as newborn brains. I encourage the authors to create a clean visualization where they only compare the CNN and transformer-based networks side by side. Again, this generalization experiment deserves another point.
> >
> > **3. Image resolution:** Thank you for clarifying that point. I have one more question: Did you use artificial data augmentations while training or testing the models? If yes, do you have results where all the artificial data augmentations were turned off before training the models? Many recent studies have shown that adding artificial image augmentation can make a model shift its feature space to learn shape-based features rather than color-based features. Can the authors comment on this point (if artificial image augmentations were used)?
> >
> > **4. Additional Details:**  The authors have addressed all of my remaining queries.
> >
> > I am satisfied with their responses, and therefore, I am willing to increase the two points that I deducted due to the lack of two major experiments.

---

> ### Author Response · Authors · 2025-08-06
> **Thank you and follow up on data augmentation**
>
> Thank you again for your feedback and additional input. We are glad the revisions addressed your concerns and appreciate your updated score.
>
> We agree that the additional analyses (e.g., egocentric models, ViTs) deserve to be clearly highlighted as individual points in the final version, and we will revise the manuscript accordingly.
>
> Regarding point 3: yes, all models we originally tested were trained with some form of data augmentation. We find your point interesting, as data augmentation is widely used during training and could significantly influence model representations. Since retraining the models on ImageNet with and without data augmentation would require substantial time, we instead examined how both effective (ED) and intrinsic dimensionality (ID) change under two conditions: i) shape-biased augmentations [1] and ii) adversarial augmentations [2] for the ResNet50 model. Interestingly, we still observe the same general increase–decrease trend in ID, suggesting that this effect is robust to these augmentation strategies. However, for i) we observe a less prominent drop of dimensionality.
>
> We plan to include a discussion on the role of data augmentation in the Discussion section. We agree that exploring how different augmentation strategies affect the dimensionality of neural representations is interesting and could also be connected to the dimensionality of datasets, as discussed in previous studies [3]. Thank you for raising this interesting point and we are happy to address any further questions or comments.
>
> **References**
>
> [1] Geirhos et al., 2022. ImageNet-trained CNNs are biased towards texture; increasing shape bias improves accuracy and robustness.
>
> [2] Engstrom et al., 2019. Robustness (Python Library).
>
> [3] Pope et al, 2021. The Intrinsic Dimension of Images and Its Impact on Learning.

---

### Official Review · Reviewer_aasq · 2025-06-23

**Clarity:** 2
**Significance:** 3
**Originality:** 3
**Rating:** 4
**Confidence:** 3

**Summary:**

This paper systematically analyzes and compares the linear and nonlinear activations of the human brain and ANN when viewing natural images. They point out that nonlinear dimensionality shows a collapse in later layers of ANNs. They found that this collapse would limit Semantic Generalization.

**Questions:**

1. Your work says “We find that in the... between brains and models” (Line 63-66). The layers of the human brain,  as hierarchical pathways, such as primary visual cortex -> intermediate visual cortex -> advanced visual cortex, while the layers of the ANN are the network layers of the model. Do these two layers mean the same thing? How do you prove that a certain layer of the model corresponds to a certain visual layer of the human brain?

2. In recent years, several viewpoints have emerged on the interpretability of ANNs. Some people believe that artificial neural networks trained in physical space are similar to the networks of real brains [1]. Others have shown that "models with higher performance on benchmark tasks achieve peak prediction accuracy in earlier layers. In contrast, models with lower performance show delayed representations and require deeper layers to achieve similar levels of brain prediction accuracy."[2]. This result means that the model is gradually becoming more and more like the brain as the level changes. When this similarity reaches its peak, it turns to generate exclusive processing, which exceeds the target range of current neural recordings, resulting in a decline in the encoding model. So, if you want to compare the levels of two "brains" in physical space, should the ANN level be truncated at the most brain-like layer (because it can best explain brain response activities)? This seems to be more objective because the deep layer will lose some interpretability due to the conversion of the target. This may get completely different results.

3. You mentioned that "ANNs exhibit a collapse in intrinsic dimensionality in deeper layers". The explanation of intrinsic dimension in the article is nonlinear correlation. If the model has a complete basis, it can well linearly represent the nonlinear components in the model's perceived information, then it will naturally get a lower intrinsic dimension because it can be well linearly approximated. What do you think about this issue?

4. You mentioned that in the brain, higher-dimensional representations are associated with stronger abstract generalization abilities. [3] seems to have made a similar conclusion. Is it necessary to discuss it?

I would be willing to improve my score if the above questions can be answered well.

**Reference**

[1] Achterberg, J., Akarca, D., Strouse, D.J. et al. Spatially embedded recurrent neural networks reveal widespread links between structural and functional neuroscience findings. Nat Mach Intell 5, 1369–1381 (2023).

[2] Mischler, G., Li, Y.A., Bickel, S. et al. Contextual feature extraction hierarchies converge in large language models and the brain. Nat Mach Intell 6, 1467–1477 (2024).

[3] Wang, A.Y., Kay, K., Naselaris, T. et al. Better models of human high-level visual cortex emerge from natural language supervision with a large and diverse dataset. Nat Mach Intell 5, 1415–1426 (2023).

**Ethical Concerns:**

["NO or VERY MINOR ethics concerns only"]

**Final Justification:**

The authors provided reasonable explanations for issues commonly raised by the reviewers, such as clarifying the relationship between brain layer stratification and ANNs, and the alignment between brain regions and model layers. Furthermore, they addressed my concerns by truncating the results at the most brain-like layers. The authors also provided reasonable responses, so I am willing to increase my score.

**Limitations:**

yes

**Quality:**

2

**Strengths And Weaknesses:**

**Strengths**

1. This paper reveals a functional mismatch in representational geometry between brains and ANNs.

2. This paper demonstrates that global average pooled features from ANNs align best with fMRI-derived linear geometry, which is a significant discovery.

**Weaknesses**

1. The conclusions of this paper are similar to those of some other literature [1], which affects its originality.

2. The explanation of the stratification of the cerebral cortex and the levels of ANN is not clear enough.

[1] Wang, A.Y., Kay, K., Naselaris, T. et al. Better models of human high-level visual cortex emerge from natural language supervision with a large and diverse dataset. Nat Mach Intell 5, 1415–1426 (2023).

---

> ### Author Rebuttal · Authors · 2025-07-30
>
> Thank you for your feedback and giving us the opportunity to clarify and improve our results. Please let us know if there is further clarification we can provide.
>
> **1. Similarity to previous studies**
>
> We thank the reviewer for highlighting this relevant study, which demonstrates that the incorporation of language can improve neural predictivity in late-stage visual regions using voxel-wise encoding models. We fully agree that this work conceptually contains some similar ideas to our work. At the same time, our study provides a complementary and distinct contribution in several key aspects. While [1] primarily focus on improving encoding model performance through training objectives, our work shifts the focus toward the representational geometry of both artificial and biological systems. To the best of our knowledge, we are the first to report a continuous effective (linear, ED) and intrinsic (nonlinear, ID) dimensionality increase along the ventral visual hierarchy in humans using large-scale fMRI data, while also comparing these patterns to ANNs. Additionally, we examine how ED and ID dimensionality evolve along the hierarchy in ANNs, and identify a consistent collapse of ID in deeper ANN layers that differs from our observations in the brain. In contrast to [1], we show that additional language information cannot resolve this dimensionality mismatch. Further, we show that deep layers with reduced ID, also show a reduced generalization to abstract, behaviorally rated features (e.g., heaviness). Importantly, these features differ from the object categories used in [1] (e.g., person, vehicle, food) by their level of abstractness and continuous values that were behaviorally determined. This adds a behavioral relevance dimension and places stronger demands on the generalization capacity of the models. Together, while both studies analyze ANN–brain alignment, our work contributes a new conceptual framework, i.e., dimensionality alignment. We agree with the suggestion to more clearly contextualize our work relative to this prior study (also related to your last question) and will revise the manuscript accordingly.
>
> **2. Clarifying the stratification of the cortex and levels of the ANN**
>
> Thank you for pointing out the lack of clarity in our explanation, a concern that was also raised by other reviewers. In our analysis, the cerebral cortex was segmented into visual hierarchy regions using the HCP atlas (see Supplement Section A1.3, [2]). We then defined cortical depth as the geodesic distance from the center of each region to the center of the V1 foveal representation, computed along the cortical surface. To ensure robustness, we additionally tested other distance measures, including Euclidean distances in both surface and volume space, and found similar results.
> In the artificial neural networks we followed previous literature to compute the relative depth [3, 4]: First, we extracted activations from all ReLU layers (see lines 116-117; GeLU layers for transformer-based models), we then normalize the position of these layers in the processing hierarchy by the number of all processing steps (layers) in the model. We will now revise the manuscript to include a more detailed explanation. Additional information regarding newer models such as ViTs will be included in the updated Supplement.
>
> **3. Alignment between brain regions and model layers**
>
> Thank you for raising this important point about the alignment between model depth and brain hierarchy. Similar concerns were also raised by other reviewers, and we have implemented three different analyses to directly address this point.
> First, we tested if the drop in dimensionality persists if we reduce our results to the layers that were identified by **BrainScore** [5] to best match V1, V2, V4, and IT. We found that both ED and ID increase up to the layers corresponding to V4, but ID notably drops in the IT layers, supporting our claim of a mismatch in representational geometry. However, as BrainScore relies on unit activations (without GAP) and is focused on macaque data, we additionally conducted two alignment analyses using the THINGS fMRI data.
> First, we trained an **encoding model to predict the vertex responses from the model’s activations** [1,5,6]. This analysis revealed that V1 is best predicted by early to intermediate layers, early processing regions (V2-V4) by intermediate layers (around peak ID), and higher ventral regions by late layers where ID has already declined, confirming that the drop in ID occurs in layers most similar to high-level brain areas,, like FFC.
> Second, to remove noise from the data we projected both fMRI activity and ANN activations onto their respective first principal components and computed the correlation. Again, we found that early and intermediate model layers align best with primary and early visual areas, and later layers with higher areas, further validating our results.
> Overall, these results suggest that the observed divergence between brains and models cannot be explained by misaligned depth and supports our conclusion that current ANNs may lack key computational properties required to maintain high-dimensional, abstract representations observed in higher-order visual cortex. We will add these findings in the revised manuscript to contextualize the models’ ID drop with respect to model–brain alignment.
>
> **4. Truncating results at the most brain-like layers**
>
> This is a valuable insight, and we thank the reviewer for highlighting recent perspectives on ANN–brain interpretability. We agree that aligning processing stages between ANNs and the brain is challenging and also depends strongly on the chosen metric [7]. Truncating results at the most brain-like layer does not qualitatively change our findings. We believe that our study adds a complementary alignment criterion: assessing whether models preserve the brain’s observed increase in representational geometry along the visual hierarchy, providing a perspective beyond voxelwise predictivity. Prior work [8] linked dimensionality reduction to improved generalization, but typically within-class generalization, not the more difficult generalization to abstract features tested here. Hence, while we agree that task-specific compression in models could contribute to the late-stage decline in dimensionality, we believe that this difference is meaningful and worth reporting. In summary, truncating ANNs at the most brain-like layer is a valid approach, but our findings show that even these layers differ structurally from biological systems. Highlighting these discrepancies is an important step toward building models that not only predict neural activity but better capture the brain’s internal organization. We will clarify this in the revised manuscript.
>
> **5. Lower nonlinear dimensionality due to a complete basis**
>
> We appreciate the comment and respond below based on our current understanding of the concern. In case we have not interpreted it correctly, we welcome further elaboration.
>
> We agree that the low intrinsic dimensionality observed in later layers could partly reflect a convergence toward a space with a complete basis, where nonlinear structures become more linearly approximable. However, given the task-specific nature of training, particularly in models like ResNet50 trained on ImageNet, we believe that the observed dimensionality collapse rather reflects a true reduction in representational diversity, potentially shaped by the model’s training objective. In the early- to mid-layers of the models, nonlinear dimensionality is behaving largely similarly to linear dimensionality, which suggests that in these layers representations capture both linear and nonlinear variations. However, in later layers the nonlinear dimensionality collapses while linear dimensionality remains high, implying that although the network maintains a set of linearly independent features, these features increasingly fail to span the richer, nonlinear structure that we observe in the human brain. This divergence may reflect a task-driven compression where only task-relevant, linearly decodable features are retained [8], limiting the model's capacity for flexible abstraction. In our work, we observe that the collapse of nonlinear dimensionality in deeper ANN layers coincides with a reduced ability to decode abstract, high-level features. While this does not establish a causal relationship, the parallel between dimensionality compression and reduced generalization suggests a potential constraint in the representational capacity of these layers. If our explanation does not yet fully capture and resolve your concerns, we would appreciate further elaboration on the issue to ensure that this point is thoroughly addressed in the revision.
>
> **References**
>
> [1] Wang et al., 2023. Better models of human high-level visual cortex emerge from natural language supervision with a large and diverse dataset.
>
> [2] Glasser et al., 2016. A multi-modal parcellation of human cerebral cortex.
>
> [3] Ansuini et al., 2019. Intrinsic dimension of data representations in deep neural networks.
>
> [4] Sorscher et al., 2022. Neural representational geometry underlies few-shot concept learning.
>
> [5] Schrimpf et al., 2018. Brain-score: Which artificial neural network for object recognition is most brain-like?.
>
> [6] Naselaris et al., 2011. Encoding and decoding in fMRI.
>
> [7] Soni et al., 2024. Conclusions about neural network to brain alignment are profoundly impacted by the similarity measure.
>
> [8] Recanatesi et al., 2019. Dimensionality compression and expansion in deep neural networks.

---

> > ### Comment · Reviewer_aasq · 2025-08-04
> >
> > I thank the authors for the response. I am in general satisfied with the update. I have raised my score.

---

> ### Author Response · Authors · 2025-08-06
> **Thank you**
>
> Thank you again for your feedback and additional input. We are glad the revisions addressed your concerns and appreciate your updated score.

---

### Official Review · Reviewer_TFdE · 2025-07-01

**Clarity:** 4
**Significance:** 3
**Originality:** 3
**Rating:** 5
**Confidence:** 4

**Summary:**

The authors probe the representational geometry of the human visual cortex and convolutional neural nets (CNNs) by measuring their effective dimensionality (ED) and intrinsic dimensionality (ID). ED measures the linear expressivity of the neural code, while ID measures the number of independent parameters needed to describe the data locally. In both human fMRI data and CNNs with global-average pooling (GAP), ED grows monotonically in deeper layers. ID grows monotonically along the human visual hierarchy but shows a hunchback profile for CNNs with GAP, indicating a mismatch. The higher ID observed along the human cortex is interpreted as higher semantic abstraction, whereas the lower ID in CNNs indicates that the final layers of deep networks lose such information. Supporting this claim, the authors show that classification tasks can be solved better with deeper layers both in the human cortex and CNNs, while abstract concepts are decoded inceasingly well only in the human data.

**Questions:**

1. How does high ID relate to disentanglement/abstraction in the sense used in ref. \[34]? Also, how does the ID defined here compare to SD in ref. \[34]?
2. How does changing $k$ affect the ID? Does setting $k = 20$ keep each term in Eq. 2 “local” when there are only 50 points in each searchlight sphere?
3. What is the importance of super-linearity, and how does it differ from mere linearity? How does this relation change if the distances are the ranks rather than geodesic distance?
4. Would you consider repeating the experiments for biologically inspired models like cornets?

**Ethical Concerns:**

["NO or VERY MINOR ethics concerns only"]

**Final Justification:**

This is an empirical paper that reports an interesting phenomenon: artificial neural networks seem to differ from biological networks in their non-linear dimensionality but not in their effective dimensionality. It is clearly written, and the experiments are thorough.

**Limitations:**

The main claim holds only for CNNs with specific pooling, thus limiting the applicability of the proposed benchmark to other architectures such as ViTs.

**Quality:**

3

**Strengths And Weaknesses:**

## Strengths

The paper was very fun to read; it is clearly written and easy to follow. Experimental methods are clearly defined with sufficient detail. Representational alignment between ANNs and brains is a hot topic. The proper metrics to measure representational geometry and interpretability are a huge debate in the NeuroAI community, and this work makes a solid contribution by providing a carefully designed and interpretable benchmark to distinguish ANNs from brains. The experiments are thorough and well-thought, e.g. testing dependence on pooling strategy, controlled experiments with randomly initialized networks, two different datasets (THINGS, BOLD), cross-validated analysis, etc.

## Weaknesses

1. I am not familiar with the searchlight approach. I could not completely understand the data pipeline and how exactly ED and ID are calculated. A short description for searchlight would be helpful for general audiance.
2.  In addition to GAP, ANN layer features are randomly pooled to 50 (out of ~512) features to compute dimensionality, but this assumes the features are uniformly distributed in a layer.
3. For ED/ID and accuracy plots, the y-axes have completely different scales between brain and ANNs. For ED, ANNs are significantly higher-dimensional and hard to compare with human data. For decoding, ANNs perform an order of magnitude better than fMRI data. The paper should at least comment on this. Also, since the paper does not compare dimensionalities between brains and ANNs, there is no need to restrict ANN dimensionality.
4. The low ID in deeper layers is explained by task specificity of ANNs. Shouldn't SimCLR have larger ID since it is self-supervised? It is not clear how the training objective affects ANN–brain alignment. Including an unsupervised model might help test the paper’s main claim.
5. It is not clear how high *non-linear* dimensionality relates to better *linear* decodability of abstract features. The authors cite ref. \[16] (line 41) for this relation, but ref. \[16] only considers linear dimensionality. Even then, the regression score cannot be explained solely by dimensionality; alignment with the task also matters ([https://arxiv.org/abs/2309.12821](https://arxiv.org/abs/2309.12821)). The authors should clarify this point.
6. Some reported values—$R^2$ in Fig. 3 and accuracy in Fig. 4—are negative and should be fixed.

---

> ### Author Rebuttal · Authors · 2025-07-30
>
> Thank you for your feedback, and we are pleased to hear that you enjoyed reading the paper. Please let us know if there is further clarification we can provide.
>
> **1. Searchlight approach**
>
> We agree that a more detailed explanation of the searchlight approach [1] will be helpful and we will include this in the revised version.
>
> **2. Distribution of features within a layer**
>
> In our analysis, we compute dimensionality based only on the feature maps (i.e., channels) after global average pooling, and not on spatially structured representations. In principle, this means that each feature map reflects a distinct, non-spatial channel, and we expect minimal correlation due to spatial proximity. Random subsampling is therefore applied to a set of relatively independent features and does not rely on any assumptions about uniform spatial distribution. We will revise the text to better convey this.
>
> **3. Differences in scale between brain and ANN dimensionality**
>
> ED, ID, and decoding accuracy are generally higher in ANNs than in brain data, partly due to spatial correlations in fMRI vertices and noise reducing decoding accuracy. Random subsampling of vertices increases brain ED values, making them more comparable to ANN results (Fig. A10). We believe subsampling ANN features remains important, as it normalizes scale differences and enables consistent comparisons of dimensionality evolution across layers. However our main claim concerns the pattern of dimensionality change across the visual hierarchy in the brain and ANNs, rather than the precise values. We will revise the manuscript to clarify these points.
>
> **4. Different ANNs with diverse training paradigms**
>
> Thank you for the suggestions to expand our analyses to other architectures and training objectives. In response, also to other reviewers, we extended our evaluation to include:
> - **Unsupervised models (Autoencoders [2]):** ID increases in the encoder but drops in the decoder, while ED remains relatively stable across all layers, revealing a different type of mismatch to the brain.
> - **Biologically inspired models (CORNets [3])**. All models show an increase in ED. For ID, CORNet-Z exhibits a continuous increase, while the recurrent models show the typical drop in late layers.
> - **Vision Transformers (ViTs, SWIN-ViT, CLIP-ViT [4-6]):** While CLIP-ViT shows the dimensionality pattern as observed in CNNs, ViTs and SWIN-ViTs show an increase-decrease pattern in ID and ED.
> - **Models trained on human-like visual diets (Ecoset, EgoVLP [7,8]):** Both models exhibit a continuous increase in ID and ED throughout the hierarchy, more closely matching trends observed in the brain.
> - **Other model architectures (AlexNet, Inception, VGG16, ResNets):** All models show a similar dimensionality behavior as ResNet50.
>
> Thanks to these additional analyses we were able to strengthen our claim by showing that the observed mismatch in ID between human visual cortex and ANNs holds across many training paradigms and architectures. Moreover, we now show that other aspects, like biological alignment or visual diet can improve the model-brain correspondence. We will include these new insights in the revised text.
>
> **5. Dimensionality reduction in SimCLR**
>
> We agree that SimCLR’s self-supervised objective might intuitively suggest higher ID in deeper layers, as it does not compress representations into class labels. However, prior work [e.g., 9] shows that contrastive models often exhibit dimensionality collapse, with highly correlated features spanning a lower-dimensional subspace. Hence, self-supervised training does not guarantee high ID. Consistent with this, both SimCLR and CLIP show the same increasing-decreasing ID pattern as classification trained models. We will clarify this point in the revised Discussion.
>
> **6. Relation between nonlinear dimensionality and linear decodability**
>
> Thank you for raising the point about the relation between nonlinear dimensionality and linear decodability. Nonlinear dimensionality reflects the intrinsic degrees of freedom in a representation, while linear decodability measures how well specific features can be extracted with a linear readout. Note that prior work has studied the capacity of representational manifolds also through the ability to linearly decode (pseudo-)classes [10, 11]. In our results, a drop in nonlinear dimensionality in deeper ANN layers correlates with reduced linear decodability of abstract features, suggesting that representations become increasingly compressed along task-relevant axes, limiting access to non-task-related features. This aligns with prior work linking dimensional increase to better generalization performance [e.g., 12]. We will clarify this connection and cite additional literature in the revised manuscript.
>
> **7. Negative $R^2$ values**
>
> We will report the $R$-values.
>
> **8. Relation between ID and shattering dimensionality**
>
> We appreciate the opportunity to clarify that while both ID and the shattering dimensionality [SD, 10] relate to complexity, they capture different aspects. ID (here estimated via MLE [13]) is a data-driven measure that relies on relative distances between data points and reflects the number of degrees of freedom in the data manifold, providing insight into the local geometric structure of the representations. In contrast, SD measures the ability of a model to separate data under all possible labelings and  “quantifies how many random classifications of a given set of points a linear readout can solve” [10]. While this can be computationally demanding, ID relies on the one-time computation of distances between close neighbors. We will clarify this distinction in the revised manuscript.
>
> **9. Effect of $k$ on the ID**
>
> Thank you for pointing out that the $k$-value, i.e., the number of neighbors used in the ID computation is an important hyperparameter that could have an effect on the results. We therefore repeated the analysis with different $k$-values ($k$=5,10, 20) and found that all results hold when changing the number of neighbors. We will include these additional analyses in the supplement.
>
> **10. Importance of super-linearity**
>
> Thank you for raising this important conceptual question. The observation of super-linear increases in both ED and ID dimensionality across the human ventral visual hierarchy initially served as a descriptive characterization of how representational complexity changes along the cortical processing stream. Super-linearity in this context might suggest a nonlinear growth in representational capacity or abstraction across stages of the hierarchy, to **potentially support a rich, abstract feature space for flexible behavior**. However, these interpretations remain speculative and are beyond the empirical scope of the current paper. While we originally included this observation to highlight the rapid increase in dimensionality in the brain and the profound difference to ANNs, our core finding is not the exact functional form of the dimensionality increase, but rather the qualitative divergence between brains and models. In light of several reviewer comments regarding also an overemphasis of this finding, **we decided to de-emphasize the importance of the precise shape of the increase** in the revised manuscript and we will adjust both the framing and language to focus on the divergence in representational geometry between brains and ANNs.
>
> **11. Different measures of cortical distance**
>
> We agree that the functional form of dimensionality increase can be affected by the type of cortical measure and order of the regions and we appreciate the suggestion of using the rank instead of the geodesic distance. We implement this suggestion by i) converting the geodesic distance into ranks instead of exact measures, and ii) using a hierarchical rank along the visual processing hierarchy [14, 15]. While the geodesic rank reveals a more linear increase in ED, ID becomes highly variable with an unclear structure. Using the hierarchical rank, we again observe a continuous increase in ED and ID, further confirming our results. Additionally, we would like to point out that in addition to the cortical geodesic distance we also tested the Euclidean distance on the cortical surface as well as in volume space. Across all of these distance metrics, we found a super-linear dimensionality increase along the ventral visual hierarchy. We will add the new results in the revised version of the manuscript.
>
> **References**
>
> [1] Chen et al., 2011. Cortical surface-based searchlight decoding.
>
> [2] Github: Horizon2333, imagenet-autoencoder.
>
> [3] Kubilius et al., 2018. Cornet: Modeling the neural mechanisms of core object recognition.
>
> [4] Dosovitskiy et al., 2020. An image is worth 16x16 words: Transformers for image recognition at scale.
>
> [5] Liu et al., 2021. Swin transformer: Hierarchical vision transformer using shifted windows.
>
> [6] Radford et al., 2021. Learning transferable visual models from natural language supervision.
>
> [7] Mehrer et al., 2021. An ecologically motivated image dataset for deep learning yields better models of human vision.
>
> [8] Lin et al., 2022, Egocentric Video-Language Pretraining.
>
> [9] Jing et al., 2021. Understanding dimensional collapse in contrastive self-supervised learning.
>
> [10] Posani et al., 2025. Rarely categorical, always high-dimensional: how the neural code changes along the cortical hierarchy.
>
> [11] Chung et al., 2018. Classification and Geometry of General Perceptual Manifolds.
>
> [12] Elmoznino & Bonner, 2024. High-performing neural network models of visual cortex benefit from high latent dimensionality.
>
> [13] Levina & Bickel, 2004. Maximum likelihood estimation of intrinsic dimension.
>
> [14] Felleman & Van Essen, 1991. Distributed hierarchical processing in the primate cerebral cortex.
>
> [15] Rolls et al., 2024. A ventromedial visual cortical ‘Where’stream to the human hippocampus for spatial scenes revealed with magnetoencephalography.

---

> > ### Comment · Reviewer_TFdE · 2025-08-04
> >
> > I thank the reviewers for addressing my concerns. I will increase my score accordingly.

---

> ### Author Response · Authors · 2025-08-06
> **Thank you**
>
> Thank you again for your feedback and additional input. We are glad the revisions addressed your concerns and appreciate your updated score.

---

### Official Review · Reviewer_6Shz · 2025-07-03

**Clarity:** 4
**Significance:** 3
**Originality:** 3
**Rating:** 5
**Confidence:** 4

**Summary:**

This submission investigates how dimensionality measures (particularly effective dimensionality and intrinsic dimensionality) vary across the fMRI responses measured from different visual regions. They compare the observed dimensionality from the fMRI responses with the dimensionality measured from the hierarchical layers of ANNs.  The paper shows that overall the dimensionality increases for deeper visual cortical regions, and features of ANNs tend to show a trend where the dimensionality begins to decrease for late layers. When using spatially averaged features, the authors observe that the trends in ANN dimensionality across layers begin to look more like the visual regions, however the ID still decreases for some late layers, suggesting a mismatch between the hierarchy of the brain and the tested ANNs.

**Questions:**

1. Is there any normalization (for instance, z-scoring) applied to the features of the neural networks or the measured fMRI activations before the dimensionality measure is computed? This preprocessing should be reported. Additionally, I would like to see the replication for Figure 1 (1) with and without z-scoring the features/fMRI responses across stimuli before dimensionality is computed. (2) with and without the per-feature/voxel mean subtracted across stimuli (but not normalizing the variance as you do in z-scoring).
I am asking about this specifically because, for the fMRI data in particular, things like the overall amplitude of the response between different voxels can dramatically change the effective dimensionality, and this type of response variability can be dramatically different from one region to another due to the SNR of the fMRI response. It thus seems particularly important to make sure that the main trends of the paper are robust to the preprocessing strategy used.

2. Is there a citation for referring to the effective dimensionality / participation ratio as the “linear dimensionality”, or is this something being used by the authors? I have not encountered it being referred to as such before.

3. The sampling for the fMRI data is performed in a searchlight approach where neighboring voxels are included in the dimensionality, while for the ANN data it is sampled by randomly choosing 50 pooled feature maps. This potentially preserves topographical organization in the fMRI data. The authors present a control experiment for this in A10 and show that the overall pattern of responses is conserved, however it looks like the absolute values of the dimensionality are much higher for the random sampling.  Could the authors explain this substantial difference?

4. Are the super-linear fits significantly better than linear fits? I am surprised that the authors are highlighting that the increase is super-linear because from looking at it, it seems to be that the V1 point (at zero) is driving this “super linear” effect. The fits with 3 points in the supplement A11 and A12 seem particularly concerning.

5. The authors talk about the importance of global-average pooling for the dimensionality alignment (201-213), where the last sentence is about them serving as a better proxy for “neural recordings”, but I think that the authors are specifically talking about alignment with fMRI data where nearby neurons are pooled? This should be clarified, and a discussion about how there might be differences when looking at individual neural responses rather than bold activity should be included somewhere in the paper.

6. The specific layers used for the deep neural network analyses and how this is mapped to the "relative depth" 0-1 scale should be more clearly explained (I couldn't find this, but maybe it is in a section of the supplement).

7. Given that the authors main interpretation of the results is that higher-order regions remain higher dimensional because they allow for behavioral generalization, it seems appropriate to discuss work on data diet and training types that encourage the deep neural networks to maintain additional information. Some work that has demonstrated changes in the training environment/task to preserve additional information and lead to better neural alignment are:
* https://openreview.net/pdf?id=AiMs8GPP5q (adds an equivariant loss and shows that this improves predictions for IT response).
* https://www.nature.com/articles/s41467-024-53147-y (shows that diversity in the visual training diet is one of the main drivers of good predictivity)
* https://journals.plos.org/plosbiology/article?id=10.1371/journal.pbio.3002366 (shows that training on multiple tasks results in the best overall predictions for audio fMRI data)

8. Please include references to the numbered supplemental figures / sections when they are referenced in the main text.

**Ethical Concerns:**

["NO or VERY MINOR ethics concerns only"]

**Final Justification:**

This paper performs a clear and well-thought-out set of analyses to examine the dimensionality of brains and computational models. The experiments are solid, and the results seem important for those who compare models with neural data (especially fMRI).

During the rebuttal, the authors addressed my concerns by stating they will make changes to the paper about the following:
- Talk about the direction of dimensionality changes rather than emphasize super-linearity
- Clarify the methods (e.g. layer depth normalization, linear dimensionality)
- Report additional control analysis (data preprocessing, brain region/model layer alignment)
- Add additional comments/discussion (searchlight vs. random sampling, fMRI vs. single units, additional references).

**Limitations:**

yes

**Quality:**

3

**Strengths And Weaknesses:**

**Strengths**

* Investigating the local dimensionality of different cortical regions on a large-scale image dataset is interesting, and seems novel to report.
* The comparisons with ANNs seem to be well thought out, and there are some critical control experiments presented in the supplement.
* The paper is well structured, and the writing and logic are clear to follow

**Weaknesses**

* The claim about super-linearity in many places of the paper seems exaggerated, as it is driven primarily by one point (primary visual cortex) and might be dependent on the way that the “cortical distance” was determined.

* The authors claim that the ID decreases for late layers which diverges from the brain, however the actual depth of the models seems somewhat arbitrary not directly aligned with the neural responses.  It is very possible that the better comparison with these visual brain regions would be only the early/mid layers of the models (where there is a steady increase in dimensionality), and higher-order brain regions not included in the analysis would actually drop in dimensionality if they were included in the fMRI analysis.

* Dimensionality measures can sometimes be dramatically dependent on the data processing, and this is often particularly true for noisy data like fMRI (for instance normalization of signals, or the noise in the data). There are some specific questions about this below.

---

> ### Author Rebuttal · Authors · 2025-07-30
>
> Thank you for your feedback. We appreciate the opportunity to improve our manuscript based on your observations. Please let us know if there is further clarification we can provide.
>
> **1. Clarifying the use of “super-linearity” in dimensionality trends and potential confounding effects of the cortical distance metric**
>
> Thank you for this thoughtful comment, which was also raised by other reviewers, and appreciate the opportunity to clarify this point. We agree that in some cases, the observed super-linear increase in dimensionality seems to be influenced by individual data points, such as the primary visual cortex. We therefore repeated these analyses with a more fine-grained partition of visual areas and in all cases but one found that a super-linear function better fits this increase than a linear function. Additionally, we would like to point out that we tested multiple methods for computing cortical distance, including geodesic and Euclidean distance on the cortical surface as well as Euclidean distance in volume space. Across all of these distance metrics, the dimensionality increase in the human cortex was consistently better fit by a super-linear function than a linear one. That said, we would like to clarify that the precise shape of the increase (i.e., whether linear or super-linear) is not the main claim we intend to emphasize. Our core observation is that both effective dimensionality (ED) and intrinsic dimensionality (ID) increase across the human visual hierarchy, whereas in ANNs, dimensionality decreases in the later layers. To reflect this more clearly, we will revise the manuscript text to **de-emphasize the notion of "super-linearity"** and instead focus on the broader, more robust contrast in the **direction of dimensionality** change between biological and artificial systems. We will incorporate these clarifications and additional analyses into the revised manuscript.
>
> **2. Alignment between brain regions and model layers**
>
> Thank you for raising this important point about the alignment between model depth and brain hierarchy. Similar concerns were also raised by other reviewers, and we have implemented three different analyses to directly address this point.
> First, we tested if the drop in dimensionality persists if we reduce our results to the layers that were identified by BrainScore [1] to best match V1, V2, V4, and IT. We found that both ED and ID increase up to the layers corresponding to V4, but ID notably drops in the IT layers, supporting our claim of a mismatch in representational geometry. However, as **BrainScore** relies on unit activations (without GAP) and is focused on macaque data, we additionally conducted two alignment analyses using the THINGS fMRI data.
> First, we trained an **encoding model to predict the vertex responses from the model’s activations** [1-3]. This analysis revealed that V1 is best predicted by early to intermediate layers, early processing regions (V2-V4) by intermediate layers (around peak ID), and higher ventral regions by late layers where ID has already declined, confirming that the drop in ID occurs in layers most similar to high-level brain areas.
> Second, to remove noise from the data we projected both fMRI activity and ANN activations onto their respective first principal components and computed the correlation. Again, we found that early and intermediate model layers align best with primary and early visual areas, and later layers with higher areas, further validating our results.
> Overall, these results suggest that the observed divergence between brains and models cannot be explained by misaligned depth and supports our conclusion that current ANNs may lack key computational properties required to maintain high-dimensional, abstract representations observed in higher-order visual cortex. We will add these findings in the revised manuscript to contextualize the models’ ID drop with respect to model–brain alignment.
>
> **3. Data preprocessing**
>
> We agree that the data preprocessing is an important aspect that might affect dimensionality. To address this, we repeated the dimensionality analyses with different preprocessing strategies: i) with and without z-scoring, and ii) with and without mean-centering. Across both fMRI and ANN data, we found that dimensionality estimates were highly correlated. ID was particularly robust, while ED showed minor variation, likely due to the impact of mean-centering and z-scoring on the covariance matrix. In contrast, ID relies on relative distances that appear less sensitive to such transformations. These robustness checks support the stability of our main conclusions, and we will include them in the revised supplement.
>
> **4. Referring to the participation ratio as linear dimensionality**
>
> We use the term linear dimensionality to describe ED, as it captures the effective number of linear dimensions in the data based on the eigenvalue spectrum of the covariance matrix. While it is not specifically common to refer to ED as linear dimensionality most studies focus on a single measure of dimensionality. To highlight the differences between ED and ID, we chose to include the terms linear and nonlinear dimensionality. We are happy to make this more clear in a revised version of the text.
>
> **5. Differences in dimensionality values between the searchlight approach and random sampling**
>
> Thank you for pointing out the differences in scale between the dimensionality values using the searchlight and the random sampling approach. Indeed, the searchlight approach in the fMRI data includes spatially contiguous vertices, which likely preserves local topographical organization, such as retinotopic mapping, that may contribute to lower observed dimensionality values in early visual areas. In contrast, the random sampling of vertices does not preserve spatial proximity and may yield higher dimensionality. We agree that this difference in sampling could contribute to the differences in absolute dimensionality values observed in Figure A10. However, our control analysis was designed to test the robustness of the relative **dimensionality trends rather than to compare the absolute values**. Importantly, although local correlation likely plays a role, the consistent increase in dimensionality across the hierarchy remains even when subsampling is applied, indicating that there are additional organizational principles beyond spatial locality driving these effects. While our primary focus will remain on the relative trends across layers and regions, which are robust across both sampling approaches, we will explicitly mention the differences in absolute values, and state that they are, to some extent, expected due to differences in spatial organization and sampling strategy.
>
> **6. Layer depth**
>
> The definition of layer depth was indeed missing from the main text and was also raised by other reviewers. In short, we followed previous literature to compute the relative depth [4,5]: First, for models with a ResNet 50 backbone, we extracted activations from all ReLU layers (GeLu activations for transformer based models), we then normalize the position of these layers in the processing hierarchy by the number of all processing steps in the model. We will add this explanation to the Methods section to improve clarity and also highlight the extracted layers by including a list of layers in the supplement.
>
> **7. Miscellaneous**
>
> Thank you for several comments regarding clarity. Indeed, our statement in lines 201–213 refers specifically to the **alignment of GAP model features with fMRI data**, and not with single-neuron recordings. We will revise the relevant paragraph to clarify this. Furthermore, we appreciate the suggestion to include a broader discussion of this distinction. We will add a note to the Discussion section acknowledging that while GAP features may align well with fMRI data, they may not be suitable proxies for electrophysiological recordings of individual neurons that may indeed best align with the direct unit activation in the models. This distinction will become increasingly relevant as advances in high-resolution fMRI methods (e.g., 7T imaging, improved coils) enable more precise measurement of neural activity across spatial and temporal dimensions, potentially enabling more direct comparisons with unit-level model activations.
>
> Thank you for the recommended **literature**, and we will include this in a brief discussion. Following a similar suggestion from another reviewer, we extended our analysis to include models trained with visual diets that better match human experiences [6,7] and indeed observed a continuous increase in the model’s ID also in the later layers. We will report these findings in the revised manuscript.
>
> Finally, we are happy to include the exact **references to the supplemental figures** and section in the main text to further improve the clarity.
>
> **References**
>
> [1] Schrimpf et al., 2018. Brain-score: Which artificial neural network for object recognition is most brain-like?
>
> [2] Wang et al., 2023. Better models of human high-level visual cortex emerge from natural language supervision with a large and diverse dataset.
>
> [3] Naselaris et al., 2011. Encoding and decoding in fMRI.
>
> [4] Alessio et al., 2019. Intrinsic dimension of data representations in deep neural networks. [5] Sorscher et al., 2022. Neural representational geometry underlies few-shot concept learning
>
> [6] Mehrer et al., 2021. An ecologically motivated image dataset for deep learning yields better models of human vision.
>
> [7] Lin et al., 2022, Egocentric Video-Language Pretraining.

---

> > ### Comment · Reviewer_6Shz · 2025-08-03
> >
> > Thank you for the clear responses. This paper seems like a solid and necessary analysis for the NeuroAI field. The proposed changes will significantly strengthen the manuscript, and I am satisfied with the responses to my questions. I was also excited to see the additional results for other reviewers evaluating additional architectures and datasets.
> >
> > I will update my score accordingly.

---

> ### Author Response · Authors · 2025-08-06
> **Thank you**
>
> Thank you again for your feedback and additional input. We are glad the revisions addressed your concerns and appreciate your updated score.

---

### Decision · Program_Chairs · 2025-09-17

**Decision:**

Accept (poster)

**Comment:**

This paper investigates how the representational dimensionality evolves across the human visual cortex compared to artificial neural networks (ANNs) trained for vision tasks. Using fMRI responses and measures of effective (linear) and intrinsic (nonlinear) dimensionality, the authors show that dimensionality increases steadily along the human ventral pathway, supporting richer semantic abstraction in higher areas. In contrast, intrinsic dimensionality collapses in later layers of ANNs, limiting semantic generalization. The work shows that this pattern is independent on the specifics of the training objective (task) when trained on the same image database.

All reviewers agreed that the work is important in the field of neuroAI, helping to understand how ANNs are similar to biological brains and how they are not. During the rebuttal phase and encouraged by the reviewers comments, the authors performed several additional analyses whose results substantially improved the significance of the work. For example, the authors extended their analysis beyond the resnet50 based class of networks (e.g., vision transformers) and for different training datasets. These additional results showed that the original findings generalize to other network architectures but that the dimensionality collapse does not happen if the networks were trained on image sets that better resemble the natural visual diet of humans (Ecoset and EgoVLP). Including these results in the final version of the paper, as promised by the authors, will further strengthen the significance of the work, demonstrating the importance of the training diet in shaping the representational geometry of both artificial and biological neural networks.

Besides the question about generalization to transformer networks and the dependence on the training set, reviewers mostly had technical comments that were all well addressed during an extensive but very constructive rebuttal process. Tellingly, by the end all reviewers had increased their original score. The paper represents a quite valuable and timely contribution to the neuroAI field.